# LARGE (VISION) LANGUAGE MODELS ARE UNSUPERVISED IN-CONTEXT LEARNERS

**Artyom Gadetsky**[*]  **Andrei Atanov**[*]  **Yulun Jiang**[*]

**Zhitong Gao**  **Ghazal Hosseini Mighan**  **Amir Zamir**  **Maria Brbić**

Swiss Federal Institute of Technology (EPFL)

## ABSTRACT

Recent advances in large language and vision-language models have enabled zero-shot inference, allowing models to solve new tasks without task-specific training. Various adaptation techniques such as prompt engineering, In-Context Learning (ICL), and supervised fine-tuning can further enhance the model's performance on a downstream task, but they require substantial manual effort to construct effective prompts or labeled examples. In this work, we introduce a *joint inference framework* for fully unsupervised adaptation, eliminating the need for manual prompt engineering and labeled examples. Unlike zero-shot inference, which makes independent predictions, the joint inference makes predictions simultaneously for all inputs in a given task. Since direct joint inference involves computationally expensive optimization, we develop efficient approximation techniques, leading to two unsupervised adaptation methods: *unsupervised fine-tuning* and *unsupervised ICL*. We demonstrate the effectiveness of our methods across diverse tasks and models, including language-only Llama-3.1 on natural language processing tasks, reasoning-oriented Qwen2.5-Math on grade school math problems, vision-language OpenFlamingo on vision tasks, and the API-only access GPT-4o model on massive multi-discipline tasks. Our experiments demonstrate substantial improvements over the standard zero-shot approach, including 39% absolute improvement on the challenging GSM8K math reasoning dataset. Remarkably, despite being fully unsupervised, our framework often performs on par with supervised approaches that rely on ground truth labels.

## 1 INTRODUCTION

Recent progress in large language and vision-language models, which we collectively refer to as foundation models (FMs), have made it possible to adapt them to solve new tasks via zero-shot inference by leveraging their general knowledge obtained during pre-training (Brown et al., 2020). For a given task, zero-shot inference obtains the prediction $y$ for an input sequence $x$ by maximizing the probability of the next token, *i.e.*, $\arg\max_y \ p(y|x)$[1]. Various methods have been proposed to enable better task adaptation, with In-Context Learning (ICL) (Brown et al., 2020; Agarwal et al., 2024; Jiang et al., 2024), fine-tuning (Hu et al., 2022; Jia et al., 2022), and prompt engineering (Wei et al., 2022b; Snell et al., 2025) emerging as the most prevalent techniques. While these methods improve upon zero-shot inference, they rely on labeled examples or require manual effort to craft effective prompts, which can pose practical limitations.

In this work, we propose a *joint inference* framework that enables fully unsupervised adaptation on a new task. Our framework generalizes the standard zero-shot inference to joint inference over $N > 1$ inputs, resulting in the following optimization problem:

$$\arg\max_{y_1,\dots y_N} p(y_1, \dots, y_N | x_1, \dots, x_N), \tag{1}$$

---

[*]Equal contribution.

[1]This usually also involves having a task-specific textual instruction which we omit here for simplicity.

where $y_1, \ldots, y_N$ are the joint predictions for the corresponding inputs $x_1, \ldots, x_N$. Compared to the zero-shot independent predictions, joint inference can guide the model to make consistent predictions and reason over multiple inputs simultaneously (Figure 1 (left)). Performing joint inference requires solving the optimization problem that is intractable for a large number of examples $N$. To address this, we develop approximation techniques, resulting in two unsupervised adaptation methods: *unsupervised fine-tuning* and *unsupervised ICL*.

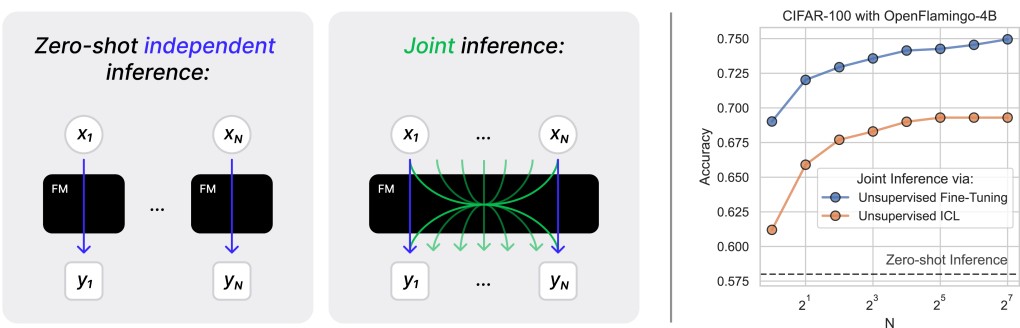

Figure 1: **Joint inference framework for foundation models.** *Left:* Unlike the standard zero-shot inference that makes a prediction $y$ independently for each input $x$, the *joint inference* makes predictions for multiple inputs at the same time, leveraging dependencies between all examples. *Right:* We develop two methods to perform the joint inference that achieve substantial improvements over traditional zero-shot inference: *unsupervised fine-tuning* and *unsupervised ICL*. Their performance increases as the number of examples $N$ for the joint inference increases, showing the effectiveness of the proposed joint inference framework.

*Unsupervised fine-tuning* is a principled method for fine-tuning an FM to optimize its own joint predictive probability (Eq. 1). The key principle lies in the self-improvement mechanism, where the FM is enhanced based on its own feedback. While this method yields strong performance, fine-tuning requires access to model weights and output probabilities, which are unavailable for close-weight models such as GPT-4 (Achiam et al., 2023). To enable the broad applicability of our joint inference framework across all model types, we introduce *unsupervised ICL*, that relies only on access to next-token generation and uses the few-shot in-context prompting to iteratively refine the predictions. Unlike supervised ICL, which uses ground truth labels for in-context examples to enhance the predictions, unsupervised ICL refines the predictions using the model's own outputs from previous iterations of refinement. We show that unsupervised ICL, in fact, implicitly maximizes the joint probability (Eq. 1) and can be seen as an approximate joint inference under the same framework.

We evaluate the proposed methods across a diverse set of tasks, including text and image classification, natural language inference, (visual) question-answering, and grade-school math problems. Our evaluation spans both language-only and vision-language FMs, including open-weight Llama-3.1 (Dubey et al., 2024), Qwen2.5-Math (Yang et al., 2024) and OpenFlamingo (Awadalla et al., 2023) models, and close-weight GPT-4o via the corresponding API. Our results show that both proposed methods significantly outperform zero-shot inference and often approach the performance of fully supervised counterparts, despite not using any labeled examples. For instance, applying unsupervised ICL to Qwen2.5-Math yields a remarkable $39\%$ absolute improvement over zero-shot inference on the GSM8K math reasoning dataset, closely matching its supervised counterpart. Similarly, unsupervised fine-tuning applied to Llama-3.1-8B, results in a substantial $23\%$ absolute improvement over zero-shot inference on average over 13 natural language processing tasks and matching the performance of the supervised fine-tuning on 6 out of 13 tasks.

## 2  RELATED WORK

**Adapting FMs via fine-tuning.** Pre-training generalist foundation models followed by task-specific fine-tunning was shown to be an effective approach to solving different language and vision tasks (Raffel et al., 2020; Radford et al., 2021; Beyer et al., 2024; Chen et al., 2023a; McKinzie et al., 2024; Dubey et al., 2024). The first pre-training stage usually involves optimizing an unsupervised

objective, *e.g.*, next-token prediction for language or contrastive loss for vision, on a large-scale dataset (Raffel et al., 2020; Cherti et al., 2023; Radford et al., 2021; Kaplan et al., 2020; Schuhmann et al., 2022). The second stage involves either full-weights training or parameter-efficient fine-tunning (Hu et al., 2022; Yosinski et al., 2014; Jia et al., 2022; Chen et al., 2023b; Houlsby et al., 2019; Pfeiffer et al., 2021). Similar to the second stage, our unsupervised fine-tuning method updates the weights of a pre-trained FM to adapt to a specific task. However, unlike other fine-tuning methods, our approach is based on a *self-improvement mechanism* and does not require labeled examples.

**Adapting FMs via prompting.** Prompting-based approaches emerged as an alternative optimization-free way to adapt an FM to a new task (Wei et al., 2022a; Radford et al., 2019; Brown et al., 2020; Alayrac et al., 2022). A standard zero-shot inference provides input and a task description as a context for a model and generates the answer via next-token prediction. A large line of works develop methods to improve this zero-shot inference by, *e.g.*, prompting a model to generate additional "reasoning" steps (Wei et al., 2022b; Yao et al., 2023; Snell et al., 2025) or providing a few labeled examples as a context (Brown et al., 2020). Similarly, our unsupervised ICL method improves upon the zero-shot inference by using a few *self-generated examples as a context* that are labeled by the model itself, thus without requiring any labeled examples.

**Reinforcement learning for FMs.** This line of work uses reinforcement learning algorithms (Sutton & Barto, 2018; Schulman et al., 2017) to fine-tune FMs to optimize a non-differentiable reward function. These reward functions are either based on a human feedback (Ouyang et al., 2022; Christiano et al., 2017), a metric (Pinto et al., 2023) or the output the same or another FM (Zheng et al., 2023; Bai et al., 2022; Lee et al., 2024). Related to this, here we *use the model's own feedback based on the joint probability* (Eq. 1) to fine-tune its weights via a reinforcement learning algorithm.

**Probabilistic Inference in FMs.** Recently, there has been a significant interest in adapting general probabilistic inference techniques to perform inference in a probabilistic models defined by a foundation model. For example, Zhao et al. (2024) build upon Sequential Monte-Carlo (Doucet et al., 2013) to sample from an unnormalized target distribution defined by a foundation model. Another line of works (Hu et al., 2024; Yu et al., 2024) employ GFlowNets framework (Bengio et al., 2023) to solve the probabilistic inference problems. While these general probabilistic inference techniques could be possibly extended to perform the proposed joint inference, we develop the principled unsupervised fine-tuning approach that effectively leverages the structure of our optimization problem.

## 3 BACKGROUND

### 3.1 FOUNDATION MODELS PRE-TRAINING

In this work, we study the class of foundation models that are pre-trained on a huge amount of data to model probabilities of a next token given the preceding ones, also known as the *next-token prediction* objective. In particular, given maximal context length $L$ of a foundation model, it models probabilities of token sequences as follows:

$$p_{\text{FM}}(t_1, \ldots, t_L) = \prod_{l=1}^{L} p_{\text{FM}}(t_l | t_{i<l}), \tag{2}$$

where $t_i \in \mathcal{V}$ and $\mathcal{V}$ is the model's vocabulary. Such pre-training has shown remarkable scaling laws (Kaplan et al., 2020), resulting in the predictable gains that can be delivered by increasing model size, the amount of available training data or compute budget. Furthermore, a separately trained vision adapter can be integrated in such models to enable performing multimodal tasks (Alayrac et al., 2022). It allows a foundation model to ingest a multimodal sequence containing images and/or videos interleaved with text and produce text.

### 3.2 ADAPTING FOUNDATION MODELS TO DOWNSTREAM TASKS

Given a pre-trained foundation model $p_{\text{FM}}$, different approaches can be used to improve the model's performance on a given task.

**Supervised fine-tuning.** Supervised fine-tuning is the prevalent approach to improve model performance on a downstream task. Specifically, given labeled examples $\mathcal{D}_{\text{train}}$, a model is trained to

maximize the probability of the correct outputs, *i.e.*, cross-entropy — $\sum_{(x, y_{\text{GT}}) \in \mathcal{D}_{\text{train}}} \log p_{\text{FM}}(y_{\text{GT}}|x)$, where $y_{\text{GT}}$ denotes the ground truth (GT) output sequence for an input $x$. Although supervised fine-tuning is the best performing, it requires access to labeled data and model weights.

**Zero-shot Inference and In-Context Learning (ICL).** Brown et al. (2020) have shown that large-scale pre-training via next-token prediction enables zero-shot inference. Specifically, without any additional training, a foundation model can be prompted with an input instance of a task $x$ and the task description $C$ to generate the corresponding solution via next-token prediction — $\arg\max_y p_{\text{FM}}(y|x, C)$. It was also demonstrated that the model's performance is susceptible to the chosen prompts, giving rise to manual prompt engineering to produce more accurate solutions (Liu et al., 2023).

Another way to improve the predictions is In-Context Learning (ICL), where a model is provided with a set of input instances and their corresponding ground truth answers. Subsequently, the model computes $\arg\max_y p_{\text{FM}}(y|x, \{(x_n, y_n^{\text{GT}})\}_{n=1}^N)$, where $(x_n, y_n^{\text{GT}})$ denote ground truth in-context examples and $N$ denotes the number of in-context examples. Although such approach has proven itself effective, it requires having access to the set of labeled examples, thus, sharing the limitations of conventional supervised learning setting.

**Chain-of-Thought (CoT).** Kojima et al. (2022) have recently proposed the off-the-shelf prompting technique that surprisingly improves the performance of a model. In particular, a model is prompted with $C_{\text{CoT}} =$ "Think step by step" phrase, that, in turn, triggers it to generate a problem solving reasoning — $r_1, \ldots, r_m \sim p_{\text{FM}}(\cdot|C_{\text{CoT}}, x)$. Subsequently, conditioning on such reasoning chain results in more accurate solutions $\arg\max_y p_{\text{FM}}(y|r_1, \ldots, r_m, C_{\text{CoT}}, x)$. The authors have also demonstrated that such approach brings improvements upon both supervised ICL and zero-shot inference.

## 4 THE JOINT INFERENCE FRAMEWORK

In this section, we first formally introduce the problem setting and then present a general form of the joint inference framework.

**Definitions and Problem Setting.** We refer to a task $\tau : \mathcal{X} \to \mathcal{Y}$ as a mapping from the space of input instances $\mathcal{X}$ to the set of plausible answers $\mathcal{Y}$. For example, for question answering, the elements of $\mathcal{X}$ and $\mathcal{Y}$ correspond to questions and the corresponding plausible answers to these questions, respectively. Another example can be the sentiment classification task, where $x \in \mathcal{X}$ are sentences, and the set of plausible answers is as simple as $\mathcal{Y} = \{\text{Positive}, \text{Negative}\}$. We assume that we are given a set of input instances $\mathcal{D} = \{x_m\}_{m=1}^M$, $x_m \in \mathcal{X}$ to perform a task $\tau$ on these instances with a foundation model $p_{\text{FM}}(\cdot)$.

The question that we aim to answer in our work is what is a principled approach to improve the predictions of $p_{\text{FM}}(\cdot)$ on a given task $\tau$ in an *unsupervised way*, *i.e.*, without having demonstrations of input instances $x$ with their corresponding correct answers $y$? To simplify the narration, we consider close-ended tasks with $K$ plausible answers, *i.e.*, $\mathcal{Y} = \{y_1, \ldots, y_K\}$, with each $y \in \mathcal{Y}$ comprising a single token. We discuss the general case of open-ended tasks in Section 6 and Appendix A.

### 4.1 GENERAL FORMULATION FOR THE JOINT INFERENCE

Here, we propose to perform joint inference to produce answers for a set of instances $\mathcal{D}$. In particular, we define the joint likelihood of $y_1, \ldots, y_M$ autoregressively given a set of instances $\mathcal{D}$ and aim to optimize the following objective:

$$\arg\max_{y_1, \ldots, y_M \in \mathcal{Y}^M} \log p(y_1, \ldots, y_M | x_1, \ldots, x_M), \text{where} \tag{3}$$

$$p(y_1, \ldots, y_M | x_1, \ldots, x_M) \stackrel{\text{def}}{=} \prod_{m=1}^M p_{\text{FM}}(y_m | x_m, \{(x_i, y_i)\}_{i<m}).$$

Given that foundation models have limited context length and processing the entire $\mathcal{D}$ might be infeasible, we consider the limited number of instances in a single model pass as follows:

$$\arg\max_{y_1,\ldots,y_M \in \mathcal{Y}^M} \mathcal{J}^N(y_1,\ldots,y_M), \text{ where} \tag{4}$$

$$\mathcal{J}^N(y_1,\ldots,y_M) \stackrel{\text{def}}{=} \mathbb{E}_{x_1,\ldots,x_N \sim \mathcal{D}} \frac{1}{N} \sum_{n=1}^{N} \log p_{\text{FM}}(y_n|x_n, \{(x_i,y_i)\}_{i<n}),$$

where $N$ limits the number of instances to be processed in a single model pass. Besides the fact that such formulation makes it possible to efficiently estimate the objective via Monte Carlo sampling, it also incorporates the important inductive bias. Indeed, Eq. (3) imposes a particular order when processing the sequence $(x_1, y_1, \ldots, x_M, y_M)$ with $p_{\text{FM}}(\cdot)$. However, ground truth answers should not depend on the particular order, and the expectation over different sequences $x_1, \ldots, x_N$ allows to effectively embed this constraint into the objective. Note that our objective is a strict generalization of the standard zero-shot inference since $\mathcal{J}^1$ reduces to it. Furthermore, Figure 1 and the results in Appendix D.1 demonstrate that increasing $N$ leads to obtaining more accurate answers for the set of instances $\mathcal{D}$ compared to the standard zero-shot inference.

Eq. (3) poses a computationally expensive combinatorial optimization problem. In the subsequent sections, we introduce two methods to optimize the proposed joint inference objective, namely, *unsupervised fine-tuning* and *unsupervised ICL*.

## 4.2 UNSUPERVISED FINE-TUNING AS A PRINCIPLED APPROACH

Although the objective $\mathcal{J}^N$ admits efficient Monte Carlo estimation, optimizing it requires $K^M M^N$ model calls which is infeasible in practice. To address this challenge, we resort to the following amortization:

$$\max_{y_1,\ldots,y_M \in \mathcal{Y}^M} \mathcal{J}^N(y_1,\ldots,y_M) \geq \max_{\theta} \mathbb{E}_{y_n \sim \tau_\theta(\cdot|x_n)} \mathcal{J}^N(y_1,\ldots,y_M), \tag{5}$$

where we refer to a $\tau_\theta(\cdot|x_n)$ as a task encoder which defines a distribution over $\mathcal{Y}$ parametrized by continuous parameters $\theta$. As a result, instead of solving the difficult combinatorial optimization problem, we can apply efficient stochastic optimization techniques to learn the parameters of the task encoder. In principle, having a flexible enough $\tau_\theta$ would result in the strict equality in Eq. (5). After the optimization is done, $\arg\max_{y \in \mathcal{Y}} \tau_\theta(y|x_n)$ provides us with answers $y_n$ to the corresponding input instance $x_n$ independently from all other input instances $x_{i \neq n}$, allowing for the efficient inference.

**Efficient optimization.** Enabling efficient optimization requires obtaining an unbiased stochastic gradient estimator of the objective in Eq. (5). However, the objective $\mathcal{J}^N$ involves evaluating $p_{\text{FM}}$ on discrete tokens $y_n$ predicted by the task encoder $\tau_\theta$, rendering this process non-differentiable. While relaxation is possible (Atanov et al., 2022; Gadetsky & Brbic, 2023; Gadetsky et al., 2024), a more prevalent approach in such scenarios is using the REINFORCE gradient estimator (Williams, 1992). Despite its generality, a naive implementation of REINFORCE suffers from the high variance when used for the optimization over combinatorial spaces (Gadetsky et al., 2020; Paulus et al., 2020; Struminsky et al., 2021). To address this challenge, we develop an effective stochastic gradient estimator that leverages the structure of our objective to substantially improve the convergence speed. We provide the complete derivation of this estimator and compare it to REINFORCE in Appendix B.1.

**Task encoder parametrization.** We employ a foundation model itself to serve as our task encoder $\tau_\theta(\cdot|x_n)$. In particular, we constrain $p_{\text{FM}}$ to model a distribution over $\mathcal{Y}$ as follows:

$$\tau_\theta(y|x_n) = \frac{p_{\text{FM}}^\theta(y|x_n) [\![y \in \mathcal{Y}]\!]}{\sum_{\hat{y} \in \mathcal{Y}} p_{\text{FM}}^\theta(\hat{y}|x_n)}, \tag{6}$$

where $[\![\cdot]\!]$ denotes Iverson bracket and $p_{\text{FM}}^\theta$ denotes the same foundation model parametrized by LoRA (Hu et al., 2022) with the corresponding trainable parameters $\theta$. The LoRA parameters $\theta$ are set such that, at the beginning of training, $p_{\text{FM}}^\theta$ corresponds to the zero-shot predictions of $p_{\text{FM}}$, providing a good initialization for our REINFORCE-based optimization, which is known to lead to faster convergence (Greensmith et al., 2004). Noteworthy, this parametrization, coupled with our

unsupervised objective, can be seen as an instantiation of self-training, in which a model improves by obtaining feedback from itself.

In principle, our unsupervised fine-tuning can be combined with any fine-tuning strategy. We provide the ablation of the task encoder's design and its effect on the performance of the unsupervised fine-tuning approach in Appendix D.2. We found that the LoRA fine-tuning provides the practical trade-off between parameter efficiency and the performance.

**Regularization.** Optimizing Eq. (5) can lead to degenerate solutions, *i.e.*, converging to a single answer for all the input instances, which is common in unsupervised learning (Gansbeke et al., 2020; Gadetsky & Brbic, 2023; Gadetsky et al., 2024; Grcic et al., 2024; Atanov et al., 2022). This happens because $p_{\text{FM}}$ assigns high probabilities to a single answer after observing the same answer for all input instances in its context. We regularize our task encoder $\tau$ to avoid such trivial solutions. Let $\tau_\theta^{\text{prior}}(y) = \mathbb{E}_{x \in \mathcal{D}} \tau_\theta(y|x)$, then the regularization term is $\mathcal{R}(\tau_\theta) = -\sum_{y \in \mathcal{Y}} \tau_\theta^{\text{prior}}(y) \log \tau_\theta^{\text{prior}}(y)$. Putting it all together, the final optimization objective to train $\tau_\theta$ is as follows:

$$\max_\theta \mathbb{E}_{y_n \sim \tau_\theta(\cdot|x_n)} \mathcal{J}^N(y_1, \ldots, y_M) + \gamma \mathcal{R}(\tau_\theta), \tag{7}$$

where we found $\gamma = 10$ is a good default choice for the regularization strength. We refer to this principled approach as the joint inference via *unsupervised fine-tuning* (Figure 2). The pseudocode and the implementation details are provided in Appendix B.1.

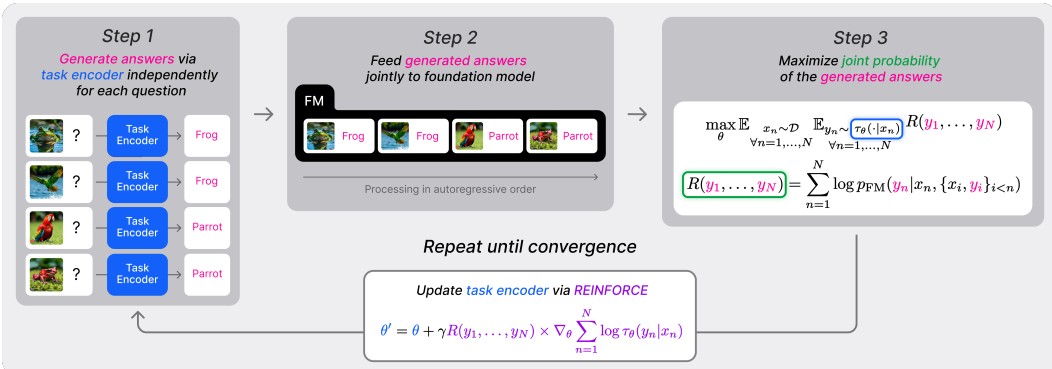

Figure 2: **Unsupervised fine-tuning** is a principled optimization method to perform *joint inference*, enabling *unsupervised adaptation* on a new task. Given a dataset of questions, each iteration of the optimization involves generating answers via task encoder independently for a batch of questions (*Step 1*). Subsequently, these answers are fed into a foundation model to estimate the joint probability, providing the quantitative measure of the quality of the answers (*Step 2*). Finally, task encoder is updated to maximize the joint probability (*Step 3*). These steps are repeated until convergence, yielding the task encoder adapted on a given task without any supervision.

## 4.3 UNSUPERVISED IN-CONTEXT LEARNING

Although amortization offers a principled approach to optimize the objective in Eq. (4), it requires access to model weights to define a task encoder $\tau_\theta$ and output probabilities $p_{\text{FM}}$. This makes unsupervised fine-tuning suitable to open-weight models, but limits its applicability to most close-weight models, such as GPT-4 (Achiam et al., 2023). To make the joint inference framework broadly applicable, our key insight is that each summand in Eq. (4) can be seen as ICL predictions:

$$\underset{y_1, \ldots, y_n \in \mathcal{Y}^n}{\arg\max} \log p_{\text{FM}}(y_n | x_n, \{(x_i, y_i)\}_{i<n}). \tag{8}$$

Unlike conventional supervised ICL, which relies on ground truth answers, our method also optimizes answers.

We employ this insight to develop the *unsupervised ICL* method to optimize Eq. (4) in the multi-turn fashion (Figure 3 (left)). Specifically, we iteratively refine answers for the set of instances $\mathcal{D}$ via conditioning on the answers from the previous round, where at the beginning they are initialized by the

zero-shot predictions. In particular, for a given $x \in \mathcal{D}$, let $y_x^0 \sim p_{\text{FM}}(\cdot|x)$ be the answers at the 0-th round. Then, for every consecutive refinement round $t$, we update the answers for $x \in \mathcal{D}$ by sampling $y_x^t \sim p_{\text{FM}}(\cdot|x, \{(x_n, y_{x_n}^{t-1})\}_{n=1}^N)$, where $x_1, \ldots, x_N \sim \mathcal{D}$. In such a way, unsupervised ICL self-improves answers through the number of iteration steps. It is important to note that this method is readily applicable to all existing foundation models, since it only requires obtaining samples from a model. Figure 3 (right) highlights that unsupervised ICL indeed optimizes the joint inference objective (Eq. (4)). We also study the effect of increasing number of turns in Appendix D.3. We provide the complete algorithm in Appendix B.2.

Furthermore, one can observe that our unsupervised ICL resembles the well-known Gibbs sampling (Geman & Geman, 1984) for sampling from untractable joint distribution by iterative sampling from tractable conditionals. Recently, similar ideas were also applied to refine CoT chains given questions and ground truth answers (Xu et al., 2024). Our unsupervised ICL pushes this idea further and, when coupled with CoT, can be seen as refining both reasoning chains and answers (Section 5.1).

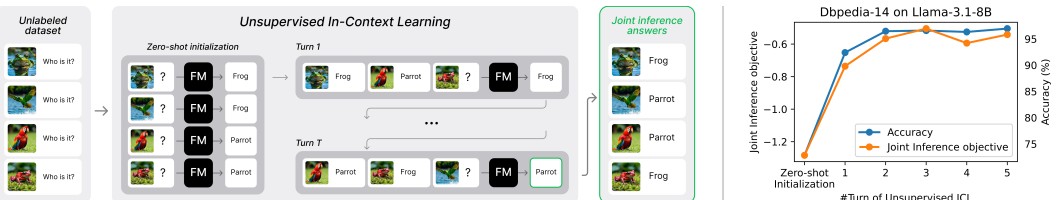

Figure 3: **Unsupervised In-Context Learning** is broadly applicable method to perform *joint inference* for *any task* and *any existing foundation model*. **Left:** Our method generates answers for each question independently using zero-shot prompting. Subsequently, it enters the multi-turn stage, where, at each turn, for each question, the model is prompted with randomly sampled in-context examples from the dataset (excluding the considered question) with the corresponding answers from the previous turn. These examples are fed into the model in the left-to-right order along with the current question to generate a refined answer. Such refinement is repeated for $T$ turns, yielding the final answers. **Right:** Both the joint inference objective and the performance improve with more optimization turns of the unsupervised ICL method.

## 5 EXPERIMENTS

**Datasets and evaluation metric.** We evaluate the performance of our two methods for joint inference across a wide range of tasks, including text classification, image classification, question answering, visual question answering, natural language inference, common-sense reasoning, and math problem-solving. A detailed description of each dataset, along with the prompts used, is provided in Appendix C.1. We use accuracy as the evaluation metric for all the experiments.

**Foundation models.** We utilize three open-source foundation models to evaluate our framework, namely, Llama-3.1 (Dubey et al., 2024) for text-based experiments, OpenFlamingo (Awadalla et al., 2023; Alayrac et al., 2022) for vision-language experiments and Qwen2.5-Math-7B (Yang et al., 2024) for reasoning experiments. Specifically, Llama-3.1-8B is used as the default model for our main text experiments, instruction-tuned version and the larger 70B instruction-tuned model are used for the ablations. For vision experiments, we use OpenFlamingo as the default model. Furthermore, we employ GPT-4o (Achiam et al., 2023) to serve us as a close-weight foundation model in our experiments.

**Baselines.** We incorporate the following baselines and upper bounds for our evaluations. (1) *Zero-shot* inference makes the predictions independently for each input example without task-specific fine-tuning or demonstrations. (2) *Zero-shot with Chain-of-Thought (CoT)* incorporates CoT reasoning prompts to generate intermediate reasoning steps before the final answer (Kojima et al., 2022; Wei et al., 2022b). We only use it for our language experiments, as we found that CoT does not show any benefit for the OpenFlamingo model, often significantly degrading the performance. (3) *Supervised In-Context Learning (ICL)* uses labeled training examples to provide them as demonstrations to the model. Consequently, this serves as an upper bound to our unsupervised ICL method, which does not use any labeled data. Similarly, (4) *Fully-Supervised Fine-tuning (FT)* employs LoRA (Hu et al., 2022) supervised fine-tuning using all labeled train-

ing examples and serves as an upper bound to our unsupervised fine-tuning method. We refer the reader to Appendix C.2 for the additional implementation details. Code is publicly available at `https://github.com/mlbio-epfl/joint-inference`.

## 5.1 RESULTS ON NATURAL LANGUAGE PROCESSING TASKS

To study the performance of the joint inference framework on language tasks, we evaluate our methods on 13 benchmark datasets, spanning various NLP tasks. Our results highlight the effectiveness of our joint inference framework (Table 1). First, the results show that unsupervised fine-tuning substantially outperforms the standard zero-shot inference. In particular, it brings 23% absolute improvement on average over 13 considered datasets, with remarkable 52.5% 30.6%, 26.3% and 19.5% on the SUBJ, Amazon, DBPedia and HellaSwag datasets, respectively. Furthermore, it often approaches the performance of its fully supervised counterpart, closely matching it on 6 out of 13 considered datasets. Secondly, unsupervised ICL also exhibits remarkable performance gains compared to the zero-shot inference, bringing 19.2% absolute improvement on average over 13 considered datasets. Remarkably, it is on par with the supervised ICL on 10 out of 13 considered datasets, overall demonstrating the effectiveness of the proposed joint inference framework.

Table 1: **Results of the unsupervised fine-tuning and unsupervised in-context learning methods on NLP tasks.** For each dataset, we show the accuracy (in %) of the zero-shot inference, the proposed unsupervised fine-tuning (FT), and ICL methods, and their corresponding supervised counterparts, which represent the upper bound. We use the Llama-3.1-8B model in all cases. Both proposed unsupervised adaptation methods outperform zero-shot inference and approach the performance of the corresponding supervised methods in most cases.

| Adaptation Method | Text Classification | | | | | | Language Inference | | | Question Answering | | | | |
|---|---|---|---|---|---|---|---|---|---|---|---|---|---|---|
| | SST2 | Amazon | AGNews | TREC | DBPedia | SUBJ | RTE | QNLI | MNLI | COPA | BoolQ | PIQA | HellaSwag | Avg. |
| Zero-shot | 77.7 | 65.5 | 74.6 | 42.7 | 72.4 | 42.9 | 62.7 | 55.5 | 34.3 | 81.0 | 66.7 | 59.0 | 46.0 | 60.1 |
| Zero-shot + CoT | 78.8 | 76.1 | 58.3 | 28.7 | 63.1 | 54.1 | 55.6 | 52.1 | 47.5 | 69.0 | 64.4 | 58.2 | 34.6 | 57.0 |
| *Fine-tunning (via LoRA):* | | | | | | | | | | | | | | |
| **Unsupervised FT** | 92.3 | 96.1 | 89.3 | 61.9 | 98.7 | 95.4 | 81.7 | 78.2 | 72.0 | 88.1 | 81.7 | 80.0 | 65.5 | 83.1 |
| Fully-Supervised | 92.1 | 96.0 | 90.4 | 93.7 | 98.8 | 96.3 | 89.0 | 89.5 | 84.7 | 85.7 | 85.6 | 82.1 | 87.1 | 90.1 |
| *In-Context Learning (no weight updates):* | | | | | | | | | | | | | | |
| **Unsupervised ICL** | 92.4 | 96.6 | 86.2 | 59.0 | 97.9 | 74.2 | 78.8 | 67.4 | 65.9 | 93.5 | 82.6 | 78.4 | 58.2 | 79.3 |
| Supervised ICL | 93.3 | 96.6 | 88.0 | 72.3 | 97.6 | 89.2 | 80.8 | 74.5 | 66.6 | 92.3 | 84.1 | 79.1 | 59.1 | 82.6 |

**Mathematical reasoning and multitask language understanding.** Furthermore, we evaluate unsupervised ICL on the GSM8K dataset that requires reasoning capabilities and on the MMLU(-Pro) dataset that covers broad knowledge of different disciplines. We employ Chain-of-Thought for our method and all the baselines on the GSM8K and MMLU-Pro datasets. Consequently, unsupervised ICL refines both reasoning chains and the answers in a fully unsupervised manner. Our results indicate that our method is also applicable to these challenging benchmarks (Table 2). For instance, it brings remarkable 39.2% absolute improvement over the zero-shot baseline on the GSM8K dataset, also outperforming the supervised counterpart.

Table 2: **Results of the unsupervised ICL on the mathematical reasoning and multiple choice question answering.** Our unsupervised ICL method improves the performance of Llama-3.1-8B and Qwen2.5-Math-7B on challenging math reasoning and multiple choice question answering.

| | Adaptation Method | GSM8K | MMLU | MMLU-Pro |
|---|---|---|---|---|
| Llama-3.1-8B | Zero-shot | 42.5 | 65.0 | 23.7 |
| | **Unsupervised ICL** | 52.8 | 66.7 | 33.1 |
| | Supervised ICL | 55.7 | 66.7 | 37.8 |
| Qwen2.5-Math-7B | Zero-shot | 52.2 | 61.0 | 37.3 |
| | **Unsupervised ICL** | 91.4 | 62.6 | 36.4 |
| | Supervised ICL | 89.9 | 62.2 | 38.7 |

**Unsupervised ICL scales effectively at test-time.** Our unsupervised ICL method consists of two stages: (1) a *task adaptation stage*, where we iteratively generate labels for unlabeled examples, creating a labeled support set; and (2) a *test stage*, where we perform ICL inference on new test examples using the labeled support set from stage one. Since the adaptation stage is performed once per task, test-time compute scaling depends on the number of (unsupervised) ICL examples used in the second test stage. Figure 4 shows how performance improves as we scale test-time compute by increasing the number of ICL examples. We find that test-time scaling of the 8B model with our method achieves a better compute-performance trade-off than zero-shot inference with the larger

70B model. Further scaling of the 70B model leads to additional gains, outperforming CoT in both compute efficiency and performance.

**Unsupervised ICL improves non-instruction-tunned models.** Figure 5 shows the impact of instruction tuning on the performance of different inference methods on the RTE dataset. We find that unsupervised ICL and FT methods with the *base model* perform better than both zero-shot and CoT with the *instruction-tuned model*, suggesting less reliance on supervised fine-tuning data. We provide further positive results of scaling to 70B model in Appendix D.4.

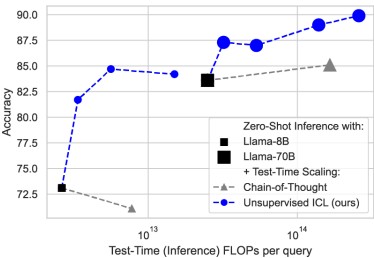

Figure 4: **Unsupervised ICL scales effectively at test-time.** We report performance on the RTE dataset with Llama-3.1 models. We scale test-time compute by using more (unsupervised) ICL examples and show it provides a better compute-performance trade-off than zero-shot inference with a bigger model.

Figure 5: **Unsupervised ICL and FT improve non-instruction-tuned models.** We show the performance of different inference methods on the RTE dataset for base and instruction-tuned Llama-8B models. Both our methods applied to the base model outperform zero-shot inference with the instruction-tuned model.

## 5.2 RESULTS FOR IMAGE CLASSIFICATION AND VISUAL QUESTION ANSWERING TASKS

To study the performance of the joint inference framework on tasks that require visual comprehension, we evaluate our methods on seven vision datasets, spanning both image classification tasks (CIFAR-10, CIFAR-100, Food101) and visual question-answering tasks (COCO-Color, COCO-Number, VQAv2 and VizWiz). Our results demonstrate that both unsupervised fine-tuning and unsupervised ICL consistently outperform the standard zero-shot inference (Table 3). In particular, unsupervised fine-tuning brings substantial absolute improvements of 14% on average over the applicable datasets with the remarkable gains of 23% on the Food101 dataset, which is the challenging fine-grained image classification task for a vision-language foundation model. Furthermore, reflecting our language experiments, unsupervised ICL closely matches the performance of its supervised counterpart on 5 out of 7 considered datasets, overall demonstrating the applicability of our joint inference framework to vision-language foundation models.

Table 3: **Results for image classification and VQA tasks**. For each dataset, we report the accuracy (in %) of zero-shot inference, the proposed unsupervised fine-tuning and unsupervised ICL, and their corresponding supervised counterparts. We use OpenFlamingo-4B in all cases except VQAv2 and VizWiz, where we use OpenFlamingo-9B. Note that unsupervised fine-tuning is not applicable to VQAv2 and VizWiz given that these datasets comprise open-ended questions. Both unsupervised fine-tuning and unsupervised ICL methods consistently outperform zero-shot inference and approach the performance of the corresponding supervised methods in most cases.

| Adaptation Method | Image Classification | | | Visual Question Answering | | | |
| --- | --- | --- | --- | --- | --- | --- | --- |
| | **CIFAR10** | **CIFAR100** | **Food101** | **COCO-Color** | **COCO-Number** | **VQAv2** | **VizWiz** |
| Zero-shot | 87.2 | 58.0 | 58.4 | 55.8 | 25.6 | 58.1 | 41.6 |
| *Fine-tunning (via LoRA):* | | | | | | | |
| **Unsupervised FT** | 96.0 | 74.1 | 81.0 | 62.0 | 42.3 | N/A | N/A |
| Fully-Supervised | 97.5 | 84.9 | 91.5 | 94.5 | 85.4 | N/A | N/A |
| *In-Context Learning (no weight updates):* | | | | | | | |
| **Unsupervised ICL** | 92.6 | 69.0 | 61.8 | 57.5 | 36.8 | 59.7 | 46.7 |
| Supervised ICL | 93.0 | 69.1 | 61.7 | 58.2 | 47.1 | 60.6 | 55.8 |

**Closed-weight GPT-4o results.** To demonstrate the applicability of the joint inference framework to closed-weight models, we employ GPT-4o and study the performance of unsupervised ICL on a subset of ImageNet, MMMU and MMMU-Pro datasets. For ImageNet, we construct a support set containing 1000 images corresponding to 100 classes and we sample 5000 images for the evaluation purposes only. Specifically, to assess the generalization, we refine the support set with our unsupervised ICL for two rounds, and, then, exam-

Table 4: **Our unsupervised ICL method improves the performance of closed-weight GPT-4o on challenging image classification and multiple choice question answering.** Note that supervised ICL is not possible as no support set is available for MMMU and MMMU-Pro datasets to perform CoT prompting.

|                  | ImageNet-100 | MMMU | MMMU-Pro |
| ---------------- | ------------ | ---- | -------- |
| Zero-shot + CoT  | 76.1         | 66.4 | 54.7     |
| **Unsupervised ICL** | 79.0     | 68.6 | 55.5     |
| Supervised ICL   | 79.5         | N/A  | N/A      |

ine the performance on the evaluation set conditioned on the refined support set. For MMMU and MMMU-Pro, we directly run methods on the validation split, since no data are available to construct support sets. Consequently, supervised ICL is not available for MMMU and MMMU-Pro. As before, we compare our unsupervised ICL to the zero-shot inference, and to the supervised ICL on ImageNet, employing ground truth labels for the support set. Table 4 illustrates that unsupervised ICL brings an improvement of $3\%$, $2\%$ and $1\%$ compared to zero-shot inference with Chain-of-Thought prompting on the ImageNet, MMMU and MMMU-Pro datasets, respectively. Furthermore, unsupervised ICL approaches supervised ICL on the ImageNet dataset, overall demonstrating that our joint inference framework is also applicable to closed-weight models.

## 6 CONCLUSION AND LIMITATIONS

In our work, we proposed the *joint inference framework* that brings substantial improvements over the standard independent zero-shot inference on a given task. The key idea behind our framework is to simultaneously make predictions for multiple input instances of a task. To perform such joint inference, which involves infeasible optimization, we develop two approximations resulting in two efficient unsupervised methods: *unsupervised fine-tuning* and *unsupervised ICL*. We show their effectiveness on a range of datasets and tasks using large language and vision-language models. Below, we discuss the framework, both methods and their corresponding limitations.

**Reliance of the joint inference framework on ICL capabilities.** In order for our joint inference framework to improve the model's zero-shot performance, the underlying model should exhibit ICL capabilities in the first place, *i.e.*, the performance of model with *supervised ICL* should be higher than its zero-shot performance. In such case, our framework allows one to invoke the model's ICL capability and improve upon the zero-shot *without the need for ground truth labels*. We study the effect of weak ICL capabilities on the performance of the joint inference framework in Appendix D.5.

**Unsupervised fine-tuning.** Unsupervised fine-tuning is a principled method to optimize the proposed joint inference objective. Remarkably, although being unsupervised, it often approaches its supervised upper bound, which uses labeled examples for fine-tuning. This method has two main limitations. First, in its current form, it is limited to close-ended tasks with a finite set of plausible answers $\mathcal{Y}$. This stems from the fact that we need to constrain the output of the task encoder to $\mathcal{Y}$, which greatly benefits the optimization. One potential solution to this could be using more advanced amortization optimization techniques such as (Hu et al., 2024; Zhao et al., 2024). Second, unsupervised fine-tuning is not applicable to closed-weight proprietary models (Achiam et al., 2023; Anil et al., 2023) since it requires access to model weights and output probabilities. We address these limitations with our unsupervised ICL method.

**Unsupervised ICL.** Unsupervised ICL offers a simple yet powerful approximation to perform the joint inference *compatible with any task and model*. It requires only obtaining samples from a model conditioned on the provided input, that is readily available for all the existing foundation models. Moreover, it can be easily coupled with modern prompting techniques such as Chain-of-Thought to further improve the performance *in an unsupervised manner*. Despite the flexibility of unsupervised ICL, similarly to how supervised ICL lags behind supervised fine-tuning, unsupervised ICL generally falls short compared to unsupervised fine-tuning. This limitation could be addressed by the improved capabilities of newly released foundation models with larger context (Agarwal et al., 2024; Jiang et al., 2024).

ACKNOWLEDGMENTS

We thank Mingqiao Ye for helping us with GPT-4o experiments. We gratefully acknowledge the support of the Swiss National Science Foundation (SNSF) grant IC00I0-231922 and the SNSF starting grant TMSGI2_226252, Zeiss and CIFAR MHU Catalyst. This work was supported as part of the Swiss AI Initiative by a grant from the Swiss National Supercomputing Centre (CSCS) under project ID a08 on Alps. This material is based on work that is partially funded by an unrestricted gift from Google.

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

## A    GENERALIZATION TO MULTI-TOKEN LABELS

In the main paper, we assume for simplicity that each $y \in \mathcal{Y}$ comprise a single token, which might not be the case for many datasets. Let $y_k = [t_1^k, \ldots, t_{l_k}^k] \in \mathcal{Y}$ be a multi-token label comprising $l_k$ tokens. Then, to compute $\log p_{\text{FM}}(y_k|x_m, \{(x_i, y_i)\}_{i<m})$ one would need to sum over all the tokens comprising $y_k$:

$$\log p_{\text{FM}}(y_k|x_m, \{(x_i, y_i)\}_{i<m}) = \sum_{i=1}^{l_k} \log p_{\text{FM}}(t_i^k|t_{j<i}^k, x_m, \{(x_i, y_i)\}_{i<m}). \tag{9}$$

Given that our task encoder $\tau_\theta$ involves renormalization in Eq. (6), summation over all the tokens for all $y \in \mathcal{Y}$ would require impractical multiple model calls.

**First token approximation.** In case of absence of labels $y \in \mathcal{Y}$ sharing their first corresponding token $t_1^k$, we found that the following approximation of Eq. (9) performs well in practice:

$$\sum_{i=1}^{l_k} \log p_{\text{FM}}(t_i^k|t_{j<i}^k, x_m, \{(x_i, y_i)\}_{i<m}) \approx \log p_{\text{FM}}(t_1^k|x_m, \{(x_i, y_i)\}_{i<m}). \tag{10}$$

**Bag-of-Tokens (BoT) approximation.** First token approximation would not work in case there are labels $y_i, y_j \in \mathcal{Y}$ that share prefix. Such scenario mostly occurs for fine-grained image classification problems. To address this challenge, we, first, find the minimal prefix $\hat{y}_k = [t_1^k, \ldots, t_{\hat{m}_k}^k], \hat{m}_k \leq m_k$ that allows to distinguish $y_k \in \mathcal{Y}$ from the rest labels. Then, we propose to consider $\hat{y}_k$ as a Bag-of-Tokens, effectively ignoring the order of $t_1^k, \ldots, t_{\hat{m}_k}^k$:

$$\sum_{i=1}^{l_k} \log p_{\text{FM}}(t_i^k|t_{j<i}^k, x_m, \{(x_i, y_i)\}_{i<m}) \approx \sum_{t \in \hat{y}_k} \log p_{\text{FM}}(t|x_m, \{(x_i, y_i)\}_{i<m}). \tag{11}$$

It is easy to note that Bag-of-Tokens approximation reduces to the first token approximation for datasets without labels that share a prefix. Consequently, we use it by default for all the datasets.

# B  IMPLEMENTATION DETAILS OF THE PROPOSED APPROACHES

## B.1  AMORTIZED APPROACH

For the close-ended tasks it is feasible to enumerate all $y \in \mathcal{Y}$, thus we renormalize conditional likelihoods over the entire set $\mathcal{Y}$, resulting in:

$$\log p(y_n|x_n, \{(x_i, y_i)\}_{i<n}) \overset{\text{def}}{=} \log \frac{p_{\text{FM}}(y_n|x_n, \{(x_i, y_i)\}_{i<n})}{\sum_{y \in \mathcal{Y}} p_{\text{FM}}(y|x_n, \{(x_i, y_i)\}_{i<n})}, \tag{12}$$

where it is important to note that this renormalization does not require additional model calls. It is well-known that rescaling the objective is beneficial for the faster convergence of REINFORCE-based optimization methods (Mnih et al., 2015; Schulman et al., 2017; Sutton & Barto, 2018). We use this renormalization for all the summands in $\mathcal{J}^N$ in Eq. (7).

**Low-variance Gradient Estimator.** Our objective in Eq. (5) has the following form:

$$\mathbb{E}_{x_1,\ldots,x_N \sim \mathcal{D}} \mathbb{E}_{y_n \sim \tau_\theta(\cdot|x_n)} \sum_{n=1}^{N} \mathcal{J}_n^N(y_1, \ldots, y_n), \text{where} \tag{13}$$

$$\mathcal{J}_n^N(y_1, \ldots, y_n) = \frac{1}{N} \log p(y_n|x_n, \{(x_i, y_i)\}_{i<n}).$$

Without loss of generality, let's consider particular samples $\hat{x}_1, \ldots, \hat{x}_N \sim \mathcal{D}$, since averaging over multiple samples does not introduce any bias. Thus, after rearranging terms, we need to obtain the unbiased gradients for the following objective:

$$\sum_{n=1}^{N} \nabla_\theta \mathbb{E}_{y_1,\ldots,y_n \sim \prod_{i=1}^{n} \tau_\theta(\cdot|\hat{x}_n)} \mathcal{J}_n^N(y_1, \ldots, y_n). \tag{14}$$

Considering only $n$-th term, let's note that:

$$\nabla_\theta \mathbb{E}_{y_1,\ldots,y_n \sim \prod_{i=1}^{n} \tau_\theta(\cdot|\hat{x}_n)} \mathcal{J}_n^N(y_1, \ldots, y_n) = \nabla_\theta \mathbb{E}_{y_1,\ldots,y_{n-1}} \sum_{y \in \mathcal{Y}} \mathcal{J}_n^N(y_1, \ldots, y_{n-1}, y) \tau_\theta(y|\hat{x}_n). \tag{15}$$

The key insight here is that marginalization over $y \in \mathcal{Y}$ can be done efficiently without additional model calls as before. Let's denote $\tilde{\mathcal{J}}(y_1, \ldots, y_{n-1}, \theta) = \sum_{y \in \mathcal{Y}} \mathcal{J}_n^N(y_1, \ldots, y_{n-1}, y) \tau_\theta(y|\hat{x}_n)$, then

$$\nabla_\theta \mathbb{E}_{y_1,\ldots,y_{n-1}} \tilde{\mathcal{J}}(y_1, \ldots, y_{n-1}, \theta) = \tag{16}$$

$$\mathbb{E}_{y_1,\ldots,y_{n-1}} \left[ \tilde{\mathcal{J}}(y_1, \ldots, y_{n-1}, \theta) \sum_{j=1}^{n-1} \nabla_\theta \log \tau_\theta(y_j|\hat{x}_j) \right] + \mathbb{E}_{y_1,\ldots,y_{n-1}} \frac{\partial}{\partial \theta} \tilde{\mathcal{J}}(y_1, \ldots, y_{n-1}, \theta),$$

where the first term can be seen as the REINFORCE gradient estimator for $\tilde{\mathcal{J}}(y_1, \ldots, y_{n-1}, \theta)$ and the second term is low-variance pathwise derivative. To reduce the variance of the overall estimator even further, we introduce simple yet effective control variate for the first term. In particular, let $y_j^* = \arg\max_{y \in \mathcal{Y}} \tau_\theta(y|\hat{x}_j), \ j = 1, \ldots, (n-1)$, then our final gradient estimator is:

$$\mathbb{E}_{y_1,\ldots,y_{n-1}} \left[ \left[ \tilde{\mathcal{J}}(y_1, \ldots, y_{n-1}, \theta) - \mathcal{B}(\hat{x}_1, \ldots, \hat{x}_n) \right] \times \left[ \sum_{j=1}^{n-1} \nabla_\theta \log \tau_\theta(y_j|\hat{x}_j) \right] \right] + \tag{17}$$

$$+ \mathbb{E}_{y_1,\ldots,y_{n-1}} \frac{\partial}{\partial \theta} \tilde{\mathcal{J}}(y_1, \ldots, y_{n-1}, \theta), \text{where}$$

$$\mathcal{B}(\hat{x}_1, \ldots, \hat{x}_n) = \sum_{y \in \mathcal{Y}} \mathcal{J}_n^N(y_1^*, \ldots, y_{n-1}^*, y) \tau_\theta(y|\hat{x}_n).$$

The obtained estimator admits the unbiased estimate by sampling $y_1, \ldots, y_N \sim \tau_\theta(\cdot|\hat{x}_n)$ and calculating what is inside expectations. Figure B1 demonstrates the effectiveness of the proposed gradient estimator on several tasks.

## B.2  MULTI-TURN FOR UNSUPERVISED IN-CONTEXT LEARNING

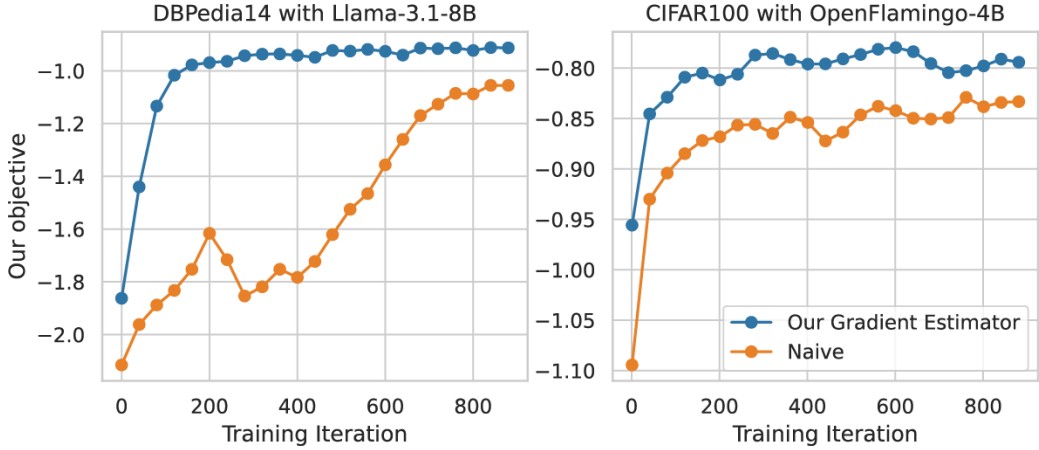

Figure B1: **Comparison of our gradient estimator with the naive approach.** The plot shows the convergence rate during optimization of the joint inference objective. Our proposed gradient estimator achieves faster convergence and leads to the higher values of the objective.

---

**Algorithm B1** Amortized Approach

---

1: **Input**: Dataset $\mathcal{D}$, Foundation model $p_{\text{FM}}(\cdot)$, hyperparameter $N$, LoRA task encoder $\tau_\theta(\cdot)$ with parameters $\theta$, regularization strength $\gamma$, number of iterations $T$, learning rate $\alpha$, batch size $B$
2: Initialize $\theta_0$ such that $\tau_{\theta_0} = p_{\text{FM}}$
3: **for** $t = 0$ to $T - 1$ **do**
4:     Sample mini-batch $x_1^b, \ldots, x_N^b \sim \mathcal{D}, \ b = 1, \ldots, B$
5:     Sample answers $y_n^b \sim \tau_{\theta_t}(\cdot | x_n^b), \ n = 1, \ldots, N; b = 1, \ldots, B$
6:     Estimate $\tau_{\theta_t}^{\text{prior}}(\cdot) = \frac{1}{N \times B} \sum_{b=1}^{B} \sum_{n=1}^{N} \tau_{\theta_t}(\cdot | x_n^b)$
7:     Compute the objective $\mathcal{O}_t = \frac{1}{B} \sum_{b=1}^{B} \sum_{n=1}^{N} \mathcal{J}_n^N(y_1, \ldots, y_n) + \gamma \mathcal{R}(\tau_{\theta_t}^{\text{prior}})$
8:     Compute the gradient estimator $g_t$ via Eq. (17)
9:     Updante the parameters: $\theta_{t+1} = \theta_t + \alpha g_t$
10: **end for**
11: Produce answers $y_n = \arg\max_{y \in \mathcal{Y}} \tau_{\theta_T}(y | x)$ for all $x \in \mathcal{D}$
12: **Output**: Answers for $\mathcal{D}$

---

**Algorithm B2** Multi-Turn Approach

---

1: **Input**: Dataset $\mathcal{D}$, Foundation model $p_{\text{FM}}(\cdot)$, hyperparameter $N$, number of turns $T$, number of repeats $N_r$
2: Initialize answers with zero-shot predictions: $\mathcal{D}_0 = \{(x, y) \mid x \in \mathcal{D}, \ y \sim p_{\text{FM}}(\cdot | x)\}$
3: **for** $t = 1$ to $T$ **do**
4:     Initialize $\mathcal{D}_t = \varnothing$
5:     **for** $x \in \mathcal{D}$ **do**
6:       **for** $n = 1$ to $N_r$ **do**
7:         Sample support examples labeled by previous turn: $(x_1, y_1^{t-1}), \ldots, (x_N, y_N^{t-1}) \sim \mathcal{D}_{t-1}$

8:         Obtain answer: $y_n^x \sim p_{\text{FM}}(\cdot | x, (x_1, y_1^{t-1}), \ldots, (x_N, y_N^{t-1}))$
9:       **end for**
10:       Take majority vote over $N_r$ options: $y^x = \text{MAJ}(y_1^x, \ldots, y_{N_r}^x)$
11:       Update answers: $\mathcal{D}_t = \mathcal{D}_t \cup \{y^x\}$
12:     **end for**
13: **end for**
14: Take answers from the last turn: $\{y_n \mid (x_n, y_n) \in \mathcal{D}_T\}$
15: **Output**: Answers for $\mathcal{D}$

---

## C  Experimental Details

### C.1  Datasets and prompts

**Text.** We evaluate our method on 16 NLP datasets covering various tasks. For sentiment analysis, we use *SST2* (Socher et al., 2013), which contains movie reviews classified as positive or negative, and *Amazon* (McAuley & Leskovec, 2013), a dataset of product reviews with similar labels. For topic classification, we use *AG-News* (Zhang et al., 2015), which consists of news articles categorized into four topics (World, Sports, Business, and Technology), *TREC* (Voorhees & Tice, 2000) for classifying questions into six types, and *DBpedia-14* (Lehmann et al., 2015), which includes Wikipedia articles grouped into 14 categories. *SUBJ* (Pang & Lee, 2004) is used for classifying sentences as subjective or objective. For natural language inference, we use *RTE* (Wang et al., 2018) to assess entailment relationships, *QNLI* (Rajpurkar et al., 2016) for sentence-answering tasks, and *MNLI* (Williams et al., 2018), which involves classifying sentence pairs into entailment, contradiction, or neutral. We also include *COPA* (Roemmele et al., 2011) and *HellaSwag* (Zellers et al., 2019) for story completion, *BoolQ* (Clark et al., 2019) for yes/no question answering, and *PIQA* (Bisk et al., 2020) for physical commonsense reasoning. For open-ended questions, *GSM8K* (Cobbe et al., 2021) assesses mathematical reasoning through multi-step word problems, *MMLU* (Hendrycks et al., 2021) and *MMLU-Pro* (Wang et al., 2024) measure multi-task language understanding and knowledge across diverse range of subjects.

For each dataset, we randomly sample $2,000$ examples as the train split for unsupervised learning, and $1,000$ examples as the test split for evaluation (except for COPA where there are only $500$ examples in total). We balance labels in both train split and test split. For GSM8K (Cobbe et al., 2021), we use the whole test set which contains $1319$ examples for the evaluation. The datasets and corresponding prompts are summarized at Table C1.

**Vision.** We evaluate our method on ten vision datasets, including four image classification tasks and six visual question-answering tasks. For image classification, we use *CIFAR10* (Krizhevsky et al., 2009), a benchmark dataset with color images across 10 different classes, *CIFAR100* (Krizhevsky et al., 2009) and *ImageNet-100* (Deng et al., 2009), which provide a more detailed classification challenge with 100 classes. We also include *Food101* (Bossard et al., 2014), a large-scale dataset featuring a wide variety of food categories. For visual question answering, we use *COCO-Color* and *COCO-Number*, both derived from VQAv2 (Goyal et al., 2018), *VQAv2* itself and *VizWiz* (Gurari et al., 2018). *COCO-Color* focuses on questions about the dominant colors of objects in images, testing the model's ability to understand color attributes, while *COCO-Number* involves predicting numerical attributes such as object counts, evaluating the model's numeric reasoning based on visual input. Furthermore, we use challenging *MMMU* (Yue et al., 2024a) and *MMMU-Pro* (Yue et al., 2024b) datasets that assess multi-discipline multimodal understanding capabilities of foundation models.

For all vision datasets, unless mentioned otherwise, we train the model on the entire training set and report performance on the test set. Details of the prompts used for each dataset can be found in Table C1.

### C.2  Implementation Details / Hyperparameters

**Unsupervised Fine-tuning.** We use LoRA (Hu et al., 2022) for parameter-efficient fine-tuning on NLP and vision tasks. For NLP tasks with Llama-3.1, we also use flash-attention (Dao et al., 2022) and 4-bit quantization of the model provided by the Unsloth library [2] to improve efficiency. We found that with improved gradient estimator, the training is less sensitive to the hyper-parameters. Thus we do not customize hyperparamters for each datasets, and instead using a learning rate of 1e-5 with Adam optimizer for all datasets. The model is fine-tuned for 6,000 iterations and usually the training converges at around 2,000 iterations. We train our model with $64$ examples at each mini-batch. We use context-length $N = 16$ for the main experiments and provide ablation study on the effect of $N$ at Appendix D.1. Similarly, for vision experiments, we train our model for 3,000 iterations with a learning rate of 1e-4, and 256 examples at each iteration. The typical training time is 12h for text tasks and 4h for vision tasks, on one NVIDIA H100 GPU.

---

[2]The library could be found at `https://github.com/unslothai/unsloth`

**Unsupervised ICL.** For unsupervised in-context learning (ICL), we initialize pseudo-labels using zero-shot predictions and iteratively refine them based on ICL predictions. At each iteration, the label of a query example is updated based on the ICL prediction from $N$ support examples. For tasks that involve reasoning process, such as GSM8K and MMLU-pro, we also initialize the reasoning chain with zero-shot Chain-of-Thought (CoT) inference and then update it through unsupervised ICL. In addition, we manually filter out unformatted responses when selecting support examples, as we find this step helpful to improve the quality of demonstrations for open-ended tasks. For both supervised and unsupervised ICL, we manually balance the labels when sampling support examples, as this helps prevent biased predictions. Additionally, we sample 5 support sequences per iteration and apply a majority vote to reduce variance (except for MMLU-pro). The labels are updated across 5 turns, if not specifically mentioned, after which we report the accuracy on the test set.

**GPT-4o evaluation.** We use the version of "gpt-4o-2024-08-06" for evaluation. We experiment on a subset of the ImageNet dataset with 1000 support images and 5000 evaluation images corresponding to 100 classes. We perform two-turn pseudo labeling for unsupervised ICL and 16-shot for evaluation. The total cost for the API call and evaluation is $200.

Table C1: Datasets and corresponding prompts used in this paper.

| Dataset | Prompts |
|---------|---------|
| SST2 | *\<sentence\>*
The sentiment of the sentence is *\<label\>*. |
| Amazon | *\<title\>\<content\>*
The sentiment of the sentence is *\<label\>*. |
| AG-News | *\<text\>*
The topic of the sentence is about *\<label\>*. |
| TREC | *\<text\>*
The topic of the sentence is about *\<label\>*. |
| DBpedia-14 | *\<title\>\<content\>*
The topic of the sentence is about *\<label\>*. |
| SUBJ | *\<text\>*
The sentence is *\<label\>*. |
| RTE | *\<premise\>*
Question: Does this imply that "*\<hypothesis\>*", yes or no?
Answer: *\<label\>*. |
| QNLI | *\<sentence\>*
Question: Does that sentence have all you need to answer the question "*\<question\>*", yes or no?
Answer: *\<label\>*. |
| MNLI | *\<premise\>*
Based on the previous passage, is it true that "*\<hypothesis\>*"?
Answer: *\<label\>*. |
| COPA | Consider the following premise: "*\<premise\>*"
Choice 1: *\<choice1\>*
Choice 2: *\<choice2\>*
Q: Which one is more likely to be the *\<question\>*, choice 1 or choice 2?
A: *\<label\>*. |
| BoolQ | *\<passage\>*
Question: After reading this passage, the answer to the question *\<question\>* is yes or no?
Answer: *\<label\>*. |
| PIQA | Goal: *\<goal\>*
Solution 1: *\<sol1\>*
Solution 2: *\<sol2\>*
Question: Given the goal, what is the correct solution, solution 1 or solution 2?
Answer: *\<label\>*. |
| HellaSwag | Consider the following description: "*\<ctx\>*"
Choice 1: *\<endings1\>*
Choice 2: *\<endings2\>*
Choice 3: *\<endings3\>*
Choice 4: *\<endings4\>*
Question: Which is the most plausible ending, choice 1, choice 2, choice 3 or choice 4?
Answer: *\<label\>*. |
| GSM8K | Given the following problem, reason and give a final answer to the problem.
Problem: *\<question\>*
Answer: *\<label\>* |
| CIFAR10 | *\<image\>*An image of *\<label\>*.*\<\|endofchunk\|\>* |
| CIFAR100 | *\<image\>*An image of *\<label\>*.*\<\|endofchunk\|\>* |
| Food101 | *\<image\>*An image of *\<label\>*.*\<\|endofchunk\|\>* |
| COCO-Color | *\<image\>*Question: *\<question\>*? Short answer: *\<label\>*\<\|endofchunk\|\>* |
| COCO-Number | *\<image\>*Question: *\<question\>*? Short answer: *\<label\>*\<\|endofchunk\|\>* |
| VQAv2 | *\<image\>*Question: *\<question\>*? Short answer: *\<label\>*\<\|endofchunk\|\>* |
| VizWiz | *\<image\>*Question: *\<question\>*? Short answer: *\<label\>*\<\|endofchunk\|\>* |
| ImageNet-100 | *\<image\>*Please identify the class of the image provided. The class has to belong to one of the classes specified in the system prompt |
| MMLU | The following are multiple choice questions (with answers) about *\<subject\>*.
*\<question\>*
Options: *\<options\>*
Answer: *\<answer\>* |
| MMLU-Pro | The following are multiple choice questions (with answers) about category. Think step by step and then finish your answer with "the answer is (X)" where X is the correct letter choice.
Question: *\<question\>*
Options: *\<options\>*
Answer: *\<answer\>* |
| MMMU,
MMMU-Pro | **Prompt for a multiple-choice question:**
*System prompt:* Answer the following multiple-choice question. The last line of your response should be of the following format: 'Answer: \$LETTER' (without quotes) where LETTER is one of the options. Think step by step before answering.
*User prompt:*
*\<question\>*
*\<options\>*
The last line of your response should be of the following format:
'Answer: \$LETTER' (without quotes) where LETTER is one of the options. Think step by step before answering.
*\<answer\>*
**Prompt for an open question:**
*System prompt:* Answer the following open-ended question. Do not use latex. Think step by step before answering. Provide the final answer in the last line of your response in the following format
'The final answer is \$ANSWER' (without quotes) where ANSWER is a single word, phrase or number.
*User prompt:*
*\<question\>*
*\<options\>*
Do not use latex. Think step by step before answering. Provide the final answer in the last line of your response in the following format 'The final answer is \$ANSWER' (without quotes) where \$ANSWER is a single word, phrase or number.
*\<answer\>* |

# D    ADDITIONAL EXPERIMENTS

## D.1    THE INFLUENCE OF THE CONTEXT LENGTH $N$

We examine the impact of the context length $N$ on the performance of our method across both language-only and vision-language tasks. As shown in Figure D1, increasing the context length consistently improves the performance for both our methods, demonstrating the benefits of the joint inference framework to improve the predictions of a foundation model upon the zero-shot inference. Remarkably, unsupervised ICL closely matches the performance of its corresponding supervised upper bound for different values of $N$. It is also worth noting that the well-known self-training principle, *e.g.* Huang et al. (2022), resembles as the special case of our unsupervised fine-tuning with $N = 1$.

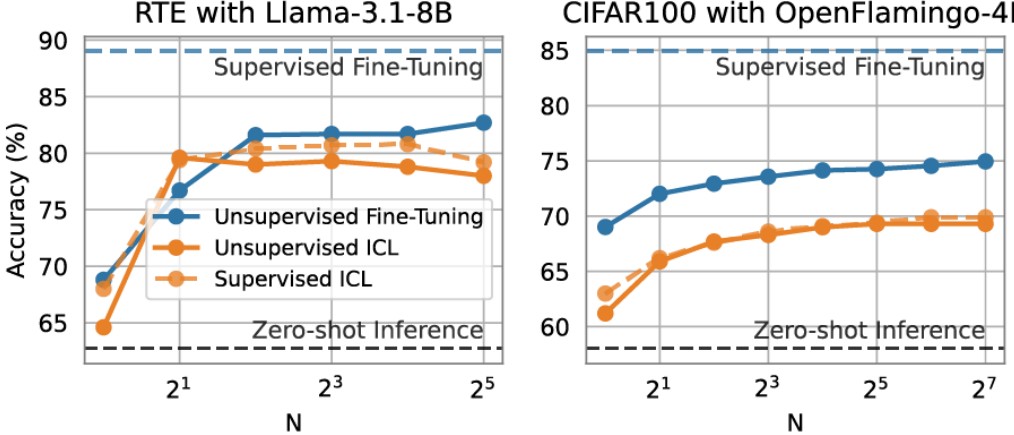

Figure D1: **The effect of the context length $N$.** We show the performance of both our methods for different context lengths ($N$). For both text (left) and image (right) classification tasks, our method displays consistent improvement as $N$ increases. This demonstrates the benefits of making joint predictions for multiple examples under the proposed joint inference framework.

## D.2    ABLATION OF TASK ENCODER PARAMETRIZATION

The choice of the parametrization of the task encoder for fine-tuning can lead to different results in terms of under-fitting and over-fitting. Notably, our method can be applied to any fine-tuning strategy, the only difference with supervised fine-tuning is that we do not need labels. We chose LoRA as a widely-adopted adaptation method that trades off over-fitting and under-fitting. To further strengthen our experimental evaluation, we conduct additional experiments of fine-tuning via *(i)* full fine-tuning and *(ii)* using a linear head on top of the fixed model (Table D1).

First, the results suggest that unsupervised fine-tuning with all three task encoders substantially outperforms the zero-shot performance on each of the considered datasets. For instance, even with the simple linear task encoder, unsupervised fine-tuning improves by 12%, 10% and 12% on the SST2, RTE and Boolq dataset respectively. On the other hand, we can observe that linear task encoder significantly underfits both LoRA and full fine-tuning. In particular, LoRA fine-tuning outperforms the linear task encoder by 3%, 9% and 3% on the SST2, RTE and Boolq datasets respectively.

Furthermore, it can be observed that supervised fine-tuning with full model training severely overfits on the RTE dataset compared to LoRA parameter efficient fine-tuning. In turn, the unsupervised fine-tuning with full model training inherits this problem and performs worse compared to the LoRA fine-tuning. In particular, compared to LoRA supervised fine-tuning, full supervised fine-tuning drops by 7% on the RTE dataset. This, in turn, results in a 2% drop of the performance of full unsupervised fine-tuning compared to LoRA unsupervised fine-tuning. Overall, the obtained results

suggest that the LoRA fine-tuning provides the practical trade-off between parameter efficiency and the performance.

Table D1: **Task encoder ablation.** LoRA fine-tuning provides the practical trade-off between parameter efficiency and the performance.

|  | Trainable Param. | SST2 | RTE | BoolQ |
|---|---|---|---|---|
| Zero-shot | - | 77.7 | 62.7 | 66.7 |
| Unsupervised FT | LoRA | 92.3 | 81.7 | 81.7 |
|  | Linear | 89.6 | 72.8 | 78.6 |
|  | Full | 92.3 | 80.3 | 81.3 |
| Supervised FT | LoRA | 92.1 | 89.0 | 85.6 |
|  | Linear | 90.1 | 74.4 | 79.9 |
|  | Full | 92.0 | 82.4 | 85.7 |

### D.3 CONVERGENCE RATE OF THE MULTI-TURN UNSUPERVISED IN-CONTEXT LEARNING

In addition, we investigate the performance of our unsupervised ICL with respect to the number of turns. The results are shown in Figure D2. Interestingly, we find that the method often converges to near-optimal performance with only a few turns, approaching the supervised ICL upper bound.

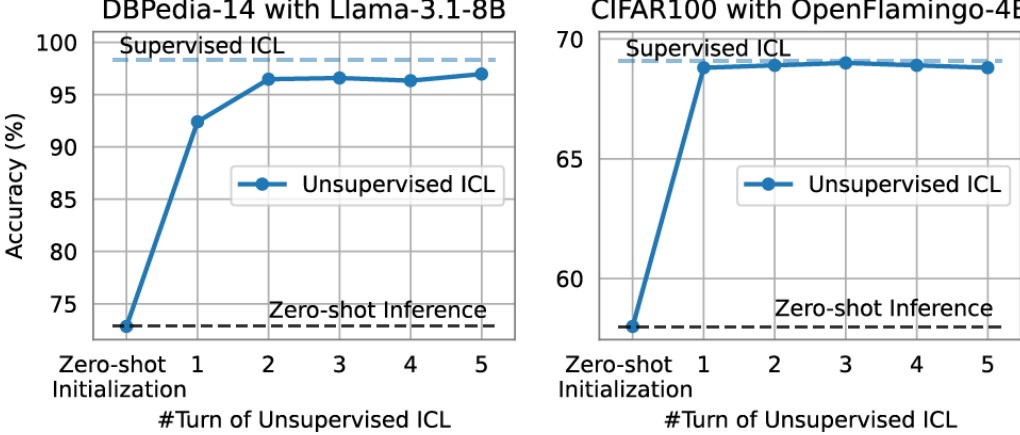

Figure D2: **The convergence analysis of the multi-turns unsupervised ICL method.** We study the number of relabeling turns need for the unsupervised ICL method to converge. We find that the proposed method converges to near-optimal performance after only a few turns and approaches the upper bound supervised ICL performance.

### D.4 THE INFLUENCE OF INSTRUCTION-TUNING AND MODEL SIZE

We study the performance of our methods, unsupervised fine-tuning and unsupervised ICL, when applied to the instruction-tuned Llama-3.1-8B and the larger scale Llama-3.1-70B models. Results show that our joint inference framework is effectively applicable across different model sizes compared to Chain-of-Thought (Figure D3), which can improve a foundation model only for large-scale models. In addition, even for the large-scale Llama-3.1-70B model, our unsupervised fine-tuning and unsupervised ICL significantly outperform the Chain-of-Thought prompting technique. In particular, it surpasses CoT by $5\%$ and $4\%$ on the SST-2 and RTE datasets, respectively, providing a principled approach to enhance predictions for models across different sizes.

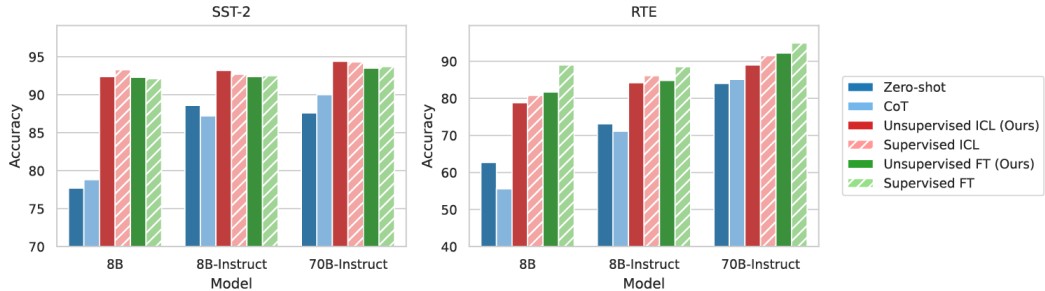

Figure D3: **Using instruction-tuned and larger scale models.** We evaluate our methods on the base 8B, instruction-tuned 8B-Instruct, and a larger 70B-Instruct from the Llama-3.1 family. We find that both proposed methods scale to instruction-tuned and larger-scale models consistently outperform zero-shot baselines. Notably, *our methods applied to the base non-tuned 8B model outperform or work closely to the zero-shot methods on a $\times 9$ larger 70B-Instruct that also benefits from additional training.*

## D.5 WEAK IN-CONTEXT LEARNING CAPABILITIES AND JOINT INFERENCE FRAMEWORK

Small models can still exhibit ICL capabilities and improve upon zero-shot performance in some cases. To show that, we conduct extensive experiments on RTE, SST2 and BoolQ datasets using small models (Table D2). We find that whenever a small model exhibits ICL capabilities, meaning that supervised ICL outperforms zero-shot baseline, one can expect improvements with the unsupervised ICL as well. For instance, on the SST2 dataset with the Phi 1.5 model supervised ICL achieves $42.7\%$ relative improvement over zero-shot and the unsupervised ICL achieves $44\%$ improvement. With the 3B parameter model, supervised and unsupervised ICL outperform zero-shot baseline by a large margin consistently across all datasets.

Table D2: **Small models can be improved via joint inference framework whenever they exhibit ICL capabilities.**

|  | RTE | | | SST2 | | | BoolQ | | |
|---|---|---|---|---|---|---|---|---|---|
|  | **Zero-shot** | **Unsup. ICL** | **Sup. ICL** | **Zero-shot** | **Unsup. ICL** | **Sup. ICL** | **Zero-shot** | **Unsup. ICL** | **Sup. ICL** |
| Llama 3.1 8B | 62.7 | 78.8 | 80.8 | 77.7 | 92.4 | 93.3 | 66.7 | 82.6 | 84.1 |
| Llama 3.2 3B | 61.6 | 73.7 | 75.0 | 72.4 | 93.3 | 92.5 | 51.3 | 75.1 | 76.2 |
| Phi 1.5 | 60.3 | 60.5 | 61.8 | 58.5 | 84.7 | 83.5 | 59.7 | 57.3 | 56.4 |
| Llama 3.2 1B | 50.8 | 55.8 | 56.5 | 68.6 | 54.3 | 61.0 | 50.0 | 50.0 | 54.9 |

Furthermore, even in cases when a small model doesn't exhibit ICL capabilities, unsupervised fine-tuning method can be used to improve this small model using feedback from a larger model with stronger ICL capabilities (Table D3). For example, for the smallest tried Llama-3.2-1B model, our unsupervised FT method significantly improves upon zero-shot when using a larger model to compute the joint inference objective and provide feedback to the small task encoder (Table D3). First, it is easy to see that even using the same 1B model to compute the joint inference objective leads to substantial gains in the performance (Table D2). For example, as shown in the experiment of previous paragraph, supervised ICL on the RTE dataset leads to only $5\%$ improvement upon the zero-shot baseline. In turn, unsupervised fine-tuning leads to $10\%$ improvement, outperforming the supervised ICL (Table D3). Furthermore, using larger models for computing the joint inference objective further improves the performance of the small Llama-3.2-1B model, closely approaching the supervised fine-tuning upper bound. For instance, on the BoolQ dataset, using the 70B model in the objective results in a remarkable $24\%$ absolute gain over the zero-shot baseline. We note that this is a one-time adaptation cost per task, after which the inference is made with the low-cost small model. Overall, these results suggest that both our methods readily applicable to a broad range of LLMs.

Table D3: **Large models that exhibit ICL capabilities can be used to improve small models with weak ICL capabilities.**

|  | **Objective compute with** | **SST2** | **RTE** | **BoolQ** |
|---|---|---|---|---|
| Zero-shot | - | 68.6 | 50.8 | 50.0 |
| Unsupervised FT | Llama 3.2 1B | 90.5 | 60.8 | 57.4 |
|  | Llama 3.2 8B | 90.8 | 74.5 | 73.4 |
|  | Llama 3.2 70B-Instruct | 90.6 | 75.6 | 74.4 |
| Supervised FT | - | 90.0 | 79.3 | 75.5 |

