# OpenReview forum: "Large (Vision) Language Models are Unsupervised In-Context Learners"
_ICLR.cc/2025/Conference — ICLR 2025 Poster_

### Official Review · Reviewer_TGs1 · 2024-10-29

**Soundness:** 2
**Presentation:** 3
**Contribution:** 3
**Rating:** 6
**Confidence:** 4

**Summary:**

This paper aims to enhance the zero-shot inference capabilities of existing open- and closed-weight models. To achieve this, it introduces a joint inference framework that processes and makes predictions for all task-related inputs simultaneously, rather than handling each input individually. The paper presents two efficient approximations—unsupervised fine-tuning and unsupervised in-context learning (ICL)—to reduce the computational complexity of joint inference. The results demonstrate the effectiveness of the proposed method across NLP and vision tasks, using two large language models (LLaMA-3 and GPT-4o) and one vision-language model (OpenFlamingo).

**Strengths:**

- I believe the idea of self-refinement using model predictions is solid, as is the approach of zero-shot evaluation without a labeled support set, which contrasts with the traditional definition of supervised in-context learning (ICL). In my view, the paper aligns well with the concept of inference-time compute scaling, as evidenced by the O1 model by OpenAI, where generating the final answer requires multiple steps (turns) by the model. Hence, the problem definition is quite sound.

- The paper presents two distinct approximations for the joint inference task, which appear to outperform all baselines. The authors demonstrate this rigorously across several NLP and vision tasks.

**Weaknesses:**

In the following, I will provide a list of weaknesses and a follow-up question for each in case the authors want to reply.

1. **Model Cacapcity and ICL Abilities**: The proposed method assumes that the pretrained model already possesses few-shot in-context learning (ICL) abilities, which are typically found in large-scale models (those with over 8 billion parameters). What about models that don't have these ICL capabilities? Can they leverage their context more effectively during joint inference? To be more specific, if the authors use the Phi-1.5 model and conduct unsupervised fine-tuning or unsupervised ICL again, what would be the outcome?

2. **Reliance on Task Encoder Approximation**: I think the design of the Task Encoder has not been explored properly in the paper. The authors only did not mention why the same pre-trained model + LoRA can be a good approximation for the task encoder approximation tasks. Why is this case? Based on which ablation in the paper? What are the other alternatives? It’s important to note that, as shown in other research, LoRA often tends to underfit a task, which can lead to suboptimal or inconsistent improvements. This raises concerns that the proposed task inference model may not accurately reflect the tasks that the model is intended to predict.

3. **Limited Applicability to Open-Ended Tasks**: Aside from the single open-ended experiment mentioned in line 395 for the GSM8K benchmark, I do not see any other experiments demonstrating the effectiveness of the proposed method for open-ended generation. With just one experiment, it is not convincing that the method is effectively generalizable for both unimodal and multimodal open-ended generation. Do the authors have any additional ablations or experiments to support the strengths of their proposed method?

4. **Missing Compute-related Ablations**: In multi-turn inference settings, it is crucial to provide the number of FLOPs used to arrive at the final answer. Do the authors have any experiments that demonstrate the scaling curve, with FLOPs on the x-axis and performance on the y-axis?

5. **Lack of Proper Evaluation in Multi-modal Setting**: In the paper, the authors used OpenFlamingo and demonstrated that the proposed method shows some effectiveness. However, I'm curious about their design choice and why they opted for OpenFlamingo. What about other alternatives that may be more powerful and capable of handling larger contexts? For example, LLaVA-Next-Interleaved is a suitable candidate, as it is trained with the in-context learning (ICL) objective function. Could the proposed method work properly if LLaVA-Next-Interleaved were used instead of OpenFlamingo? Additionally, regarding multimodal evaluation, assessing a model using COCO-color and COCO-count might not provide an accurate reflection of the proposed method's performance. Have the authors considered evaluation on the MMMU Benchmark? This benchmark is recognized as a legitimate evaluation tool for multimodal large language models, and LLaVA-Next-Interleaved has reported strong results on it.

**Questions:**

Asked above.

---

> ### Author Response · Authors · 2024-11-22
>
> We thank the reviewer for the detailed and thoughtful feedback. We are glad that the reviewer recognizes our joint inference framework and appreciates the soundness of our idea to perform unsupervised adaptation of foundation models. We are also glad that the reviewer finds our paper well-aligned with the recent trends to scale inference-time compute to improve the predictions of foundation models. Finally, we are happy that the reviewer appraises our experiments as rigorous, demonstrating the significant gains brought by the proposed approaches across various models and tasks. Below, we provide a detailed response to the reviewer’s concerns, and we hope that the provided clarifications, along with the additional experiments, will increase the reviewer’s confidence in our work.
>
> >Model Capacity and ICL Abilities: The proposed method assumes that the pre-trained model already possesses few-shot in-context learning (ICL) abilities, which are typically found in large-scale models (those with over 8 billion parameters). What about models that don't have these ICL capabilities? Can they leverage their context more effectively during joint inference? To be more specific, if the authors use the Phi-1.5 model and conduct unsupervised fine-tuning or unsupervised ICL again, what would be the outcome?
>
> We kindly refer the reviewer to the general response for the discussion on the reliance on the ICL capabilities. Similar to methods like CoT [1, 2], which relies on the model’s reasoning capabilities, our method relies on the ICL capabilities to improve upon 0-shot performance.
>
> We also provide additional results for smaller models. We make two observations:
> 1. Small models can still exhibit ICL capabilities and improve upon 0-shot in some cases. To show that, we conduct extensive experiments on RTE, SST2 and BoolQ datasets using small models (table below). We find that whenever a small model exhibits ICL capabilities meaning that supervised ICL outperforms zero-shot baseline, one can expect improvements with the unsupervised ICL as well. For instance, on the SST2 dataset with the Phi 1.5 model supervised ICL achieves 42.7% relative improvement over zero-shot and the unsupervised ICL achieves 44% improvement. With the 3B parameter model, supervised and unsupervised ICL outperform zero-shot baseline by a large margin consistently across all datasets.
>
> __RTE__
> || 0-shot | unsupervised ICL | supervised ICL |
> |-|-|-|-|
> | Llama 3.1 8B | 62.7   | 78.8             | 80.8           |
> | Llama 3.2 3B | 61.6   | 73.7             | 75.0           |
> | Phi 1.5      | 60.3   | 60.5             | 61.8           |
> | Llama 3.2 1B | 50.8   | 55.8             | 56.5           |
>
> __SST2__
> | | 0-shot | unsupervised ICL | supervised ICL |
> |-|-|-|-|
> | Llama 3.1 8B | 77.7   | 92.4| 93.3|
> | Llama 3.2 3B | 72.4   | 93.3| 92.5|
> | Phi 1.5      | 58.5   | 84.7| 83.5|
> | Llama 3.2 1B | 68.6   | 54.3| 61.0|
>
> __BoolQ__
>
> | | 0-shot | unsupervised ICL | supervised ICL |
> |-|-|-|-|
> | Llama 3.1 8B | 66.7   | 82.6| 84.1|
> | Llama 3.2 3B | 51.3   | 75.1| 76.2|
> | Phi 1.5      | 59.7   | 57.3| 56.4|
> | Llama 3.2 1B | 50.0   | 50.0| 54.9|
>
> 2. For the smallest tried Llama-3.2-1B model, our unsupervised FT method significantly improves upon 0-shot when using a larger model to compute the joint inference objective and provide feedback to the small task encoder (table below). First, it is easy to see that even using the same 1B model to compute the joint inference objective leads to substantial gains in the performance (table below). For example, as shown in the experiment of the previous response, supervised ICL on the RTE dataset leads to only 5% improvement upon the zero-shot baseline. In turn, unsupervised fine-tuning leads to 10% improvement, outperforming the supervised ICL (table below).  Furthermore, using larger models for computing the joint inference objective further improves the performance of the small Llama-3.2-1B model, closely approaching the supervised fine-tuning upper bound. For instance, on the BoolQ dataset, using the 70B model in the objective results in a remarkable 24% absolute gain over the zero-shot baseline. We note that this is a one-time adaptation cost per task, after which the inference is made with the low-cost small model.
>
> __Unsupervised FT for Llama-3.2-1B model__
> | Method     | Objective computed with| SST2 | RTE  | BoolQ |
> |-|-|-|-|-|
> | Zero-shot  | -| 68.6 | 50.8 | 50.0  |
> | Unsup FT   | Llama 3.2 1B| 90.5 | 60.8 | 57.4  |
> || Llama 3.2 8B| 90.8 | 74.5 | 73.4  |
> || Llama 3.2 70B-Instruct | 90.6 | 75.6 | 74.4  |
> | Sup FT     | -| 90.0 | 79.3 | 75.5  |
>
> Again, we thank the reviewer for the insightful suggestion to perform the additional analysis, which we will include in future revisions of our manuscript.
>
> [1] Chain-of-Thought Prompting Elicits Reasoning in Large Language Models. Wei et al. NeurIPS 2022.
>
> [2] Large Language Models are Zero-Shot Reasoners. Kojima et al. NeurIPS 2022.

---

> ### Author Response · Authors · 2024-11-22
>
> > Reliance on Task Encoder Approximation: I think the design of the Task Encoder has not been explored properly in the paper. The authors only did not mention why the same pre-trained model + LoRA can be a good approximation for the task encoder approximation tasks. Why is this case? Based on which ablation in the paper? What are the other alternatives? It’s important to note that, as shown in other research, LoRA often tends to underfit a task, which can lead to suboptimal or inconsistent improvements. This raises concerns that the proposed task inference model may not accurately reflect the tasks that the model is intended to predict.
>
> Indeed, the choice of the parametrization for fine-tuning can lead to different results in terms of under- and over-fitting. Note, that our method can be applied to any fine-tuning strategy, the only difference with supervised fine-tuning is that we do not need labels. We chose LoRA  as a widely-adopted adaptation method that trades off over- and under-fitting. To further strengthen our experimental evaluation, we conduct additional experiments of fine-tuning via 1) full fine-tuning and 2) using a linear head on top of the fixed model. We find that LoRA results in overall the best performance for both supervised and proposed unsupervised fine-tuning.
>
> We thank the reviewer, and we will include these additional experiments in the final version.
>
> |                 | Trainable Param. | SST2 |  RTE | BoolQ |
> |-----------------|:----------------:|:----:|:----:|:-----:|
> | Zero-Shot       |         -        | 77.7 | 62.7 |  66.7 |
> | Unsupervised FT |       LoRA       | 92.3 | 81.7 |  81.7 |
> |                 |      Linear      | 89.6 | 72.8 |  78.6 |
> |                 |        Full      | 92.3 | 80.3 |  81.3 |
> | Supervised FT   |       LoRA       | 92.1 | 89.0 |  85.6 |
> |                 |      Linear      | 90.1 | 74.4 |  79.9 |
> |                 |        Full      | 92.0 | 82.4 |  85.7 |
>
>
> > __Limited Applicability to Open-Ended Tasks__ & __Lack of Proper Evaluation in Multi-modal Setting:__
>
> We kindly refer the reviewer to the general response for the additional experiments on more recent benchmarks: MMMU, MMMU-Pro, VQAv2, and VizWiz for VLMs and MMLU for LLMs. We show that our method can be applied to them and improve upon the zero-shot baseline. For instance, on the MMLU benchmark, unsupervised ICL brings 3% improvements upon the zero-shot and remarkably matches the performance of supervised ICL. Furthermore, on the challenging MMLU-Pro, our unsupervised ICL improves by 6% upon the zero-shot counterpart. We also observe similar trends on the newly added vision tasks. Indeed, our unsupervised ICL brings 2% absolute improvement upon the zero-shot baseline for the frontier GPT-4o mode on the MMMU datasetl. Moreover, on the challenging VizWiz dataset, our unsupervised ICL achieves 5% absolute improvement upon the zero-shot baseline. Overall, these results highlight the strength of the proposed joint inference framework and its applicability to the frontier models and the most recent challenging benchmarks.
>
> In addition, in response to the reviewer’s request, we have also evaluated the LLaVA-Next-Interleave-7B model on MMMU and MMMU-Pro and observed that this model does not exhibit supervised ICL capabilities on the aforementioned benchmarks. In particular, zero-shot inference results in 40.7% and supervised ICL results in 38.4% on the MMMU dataset. On the MMMU-Pro dataset, zero-shot inference results in 22.1% and supervised ICL results in 19.3%. Therefore, we decided to take Qwen2-VL-72B, the most capable open-source model on this benchmark [1] that can be triggered to generate intermediate reasoning before answering the question. The zero-shot inference on the MMMU-Pro Standard dataset resulted in 49.7%, and our unsupervised ICL achieved 50.6%, improving the best result for the open-source model on this benchmark.
>
> [1] https://mmmu-benchmark.github.io/#leaderboard

---

> ### Author Response · Authors · 2024-11-22
>
> >Missing Compute-related Ablations: In multi-turn inference settings, it is crucial to provide the number of FLOPs used to arrive at the final answer. Do the authors have any experiments that demonstrate the scaling curve, with FLOPs on the x-axis and performance on the y-axis?
>
> We thank the reviewer for raising a relevant and interesting point. Below, we discuss how our method can be scaled at test time and provide the scaling curves.
>
> Our unsupervised ICL method can be split into two stages.
> 1. The first is the _task adaptation stage_. At this stage, we perform the described multi-turn optimization procedure to obtain labels for a set of unlabelled examples. This results in a labeled support set for ICL inference in the second stage.
> 2. The second stage is actually _the test stage_, where we make predictions for new incoming test examples. At this test stage, for each test example, we perform ICL inference using examples with labels found in the first stage as in-context examples.
>
> The cost of the first stage is fixed and single-time, as we do it once for a given task and do not need to do any additional turns on the obtained support set. __Therefore, our test time compute scaling depends only on the number of ICL shots used to make predictions for each test example during the second testing stage__, as a longer context requires more FLOPs.
>
> __Figure:__ https://imgur.com/a/wzS0QFs
>
> In the attached figure, we show how the performance of the unsupervised ICL changes as we scale the test time compute by increasing the number of ICL examples (blue curves). We perform the experiment for Llama-Instruct 8B and 70B and the RTE dataset. __We show that test time compute scaling of the 8B model with our method achieves a better compute-performance trade-off than zero-shot inference with a larger 70B model__. Further scaling test time compute with the 70B model further improves the performance. In all cases, we outperform the CoT approach both in terms of compute and performance.
>
> We thank the reviewer, and we will include these results in the final version of our manuscript.

---

> > ### Comment · Reviewer_TGs1 · 2024-11-24
> > **Response to Authors' rebuttal**
> >
> > Dear Authors, I appreciate your efforts to address all of my questions and concerns. I have truly enjoyed reading your rebuttal, and I would now like to increase my score from "5" to "6".

---

> > > ### Author Response · Authors · 2024-11-24
> > >
> > > We thank the reviewer for the response and for increasing the score. We are happy that all the reviewer’s concerns have been resolved by our rebuttal. We are grateful for the insightful feedback suggested by the reviewer and we will incorporate new results in the final version of our manuscript.

---

### Official Review · Reviewer_Nhh8 · 2024-11-03

**Soundness:** 3
**Presentation:** 1
**Contribution:** 3
**Rating:** 6
**Confidence:** 3

**Summary:**

The paper introduces an unsupervised joint inference framework to improve the performance of LLMs and VLMs by considering the entire set of questions for a given task instead of making predictions on each question independently. They utilize the shared context of these questions to improve model accuracy It includes a self-improvement loop where the model iteratively refines its predictions by using its own prior answers as in-context examples to guide the next round of predictions. This self-refinement continues over multiple iterations, allowing the model to gradually improve and make more consistent predictions without needing labeled data. The authors show that this

**Strengths:**

- the motivation of utilizing the entire corpus of the task to improve overall performance is compelling, as there is often shared context that is underutilized
- this method is versatile and applies to both open and closed source models as well as across modalities
- the results indicate that this can provide substancial improvements over zero shot methods

**Weaknesses:**

- the main weakness of this paper is clarity: sections 3 and 4 are hard to follow and it would have been significantly clearer to provide a methods figure which outlines the iterations and IC examples used. If I understand this method correctly, it uses responses to questions from previous iterations as in context examples for questions in the current iteration, optimizing for the joint likelyhood over all the examples in the batch. If this is the case, then it should have been stated clearly in prose in section 4. I find the entirety of section 3 to be unnecessary as it is explaining commonly known procedures for model training and inference, and much of section 4 to use unnecessary notation in explaining a relatively straightforward method. Again, I have no major concerns about the method itself, but I urge the authors to remove or significantly shorten section 3, add a methods figure, and rewrite section 4. A methods figure could be in the form of showing the inputs to the LLM at each iteration for the no weight-update option to indicate the change in in context examples.
- I am not fully convinced that adding iterations helps for the no weight update setting. if the authors added a 1-iteration baseline to their results this would resolve my concern
- all the benchmarks used in the results section are out of date and focus on easy tasks. The results would be much more compelling if the authors included a benchmark like MMLU. That being said, given the short rebuttal period I would put this experiment as a lower priority than the two above concerns

I acknowledge that I may have misinterpreted the method so some of these critiques could be unfounded. If that's the case, I am happy to learn more about the method - preferably via a methods figure - in the rebuttal and change my rating

**Questions:**

- how does your method answer multiple questions with one model call? What is the exact prompt used and how are the in context examples formatted? This should be in the appendix, table C1 does not indicate where in context examples are placed nor does is show how multiple questions are answered in a single pass. If the authors could give a template of the prompt used in the iteration steps for a benchmark that would be sufficient
- are the in context examples used different than the questions the LLM is tasked to answer in a given call?

---

> ### Author Response · Authors · 2024-11-22
>
> We thank the reviewer for the valuable feedback and for recognizing the substantial improvements brought by our method over the zero-shot baselines. We are also glad that the reviewer finds our framework compelling and the proposed approaches broadly applicable to both open and closed source models as well as across modalities. Below, we address the reviewer's concerns and we hope that our response helps to increase the reviewer's confidence in our work.
>
> > the main weakness of this paper is clarity…
>
> &
>
> > Q1: how does your method answer multiple questions with one model call? What is the exact prompt used and how are the in context examples formatted?
>
> We thank the reviewer for the fruitful suggestions to improve the presentation of our joint inference and the descriptions of the proposed approaches. Table C1 provides the template for a single pair of input instance (in blue) and answer (in red). Although both the proposed approaches do not generate answers for multiple questions with a single model call, they optimize the joint inference objective that, intuitively, corresponds to such behavior.
>
> ## Unsupervised ICL
>
> In response to the reviewer’s request and to further elaborate on the details, we provide the method figure for the unsupervised ICL approach (New-Fig. 1: https://imgur.com/a/fNnoKNs), which will be included in the final manuscript.
> As New-Fig. 1 shows, first, our method generates answers for each test example using zero-shot prompting. Subsequently, the method enters the multi-turn optimization phase, where, at each turn, for each example in the test dataset, the model is prompted with randomly sampled in-context examples from the dataset (excluding the test example) using labels from the previous iteration. These examples are fed into the model in the left-to-right order along with the current query input instance to generate a refined answer. In other words, we concatenate the templates for all the pairs with a query instance and feed them to the model. This algorithm can also be seen in Algorithm B2 in L955. As shown in Figure 2 in the paper, this method indeed optimizes the joint inference objective.
>
> ## Unsupervised FT
> In the unsupervised fine-tuning method, at each optimization step, first, the task encoder $\tau_\theta$ generates the answers for each input instance independently $y_n \sim \tau_\theta(x_n)$. Then, given predicted $y$s, parameters $\theta$ are optimized to maximize the joint inference objective $p_{\mathrm{FM}}(y_1, \dots, y_N | x_1, \dots, x_N)$, as defined in Eq. (7) and (8). This corresponds to the optimization problem in Eq. 12.
> Note that while the task-encoder makes predictions independently, after optimization with the joint objective, it approximates the result of the joint inference as shown in Sec. 4.2 and Eq. (9).
>
> We will include the new method figure and significantly improve the clarity and presentation of both methods in the camera ready version of our manuscript. We will remove unnecessary details and notation and make sure that our method is explained in a clear and concise way using the newly made image.
>
> > I am not fully convinced that adding iterations helps for the no weight update setting. if the authors added a 1-iteration baseline to their results this would resolve my concern
>
> Our unsupervised ICL method can be viewed as an iterative optimization approach to maximize the joint inference objective. We note, therefore, that the **“1-iteration baseline” also corresponds to our proposed unsupervised ICL method** run only for 1 optimization iteration (turn), as it requires ICL inference with labels obtained during 0th iteration (zero-shot initialization). In the main paper, we study the performance of our method when increasing the number of iterations, including 1-iteration. Specifically, **Fig. 2 shows that, generally, having more iterations results in a higher performance**.
>
> Akin to any other optimization method, the convergence speed of our multi-iteration optimization method can vary between different datasets. For example, Fig. 5 shows that for the CIFAR-100 dataset, one step is enough to converge. In such cases, this means that our method is able to converge fast and outperform the zero-shot baseline (the left-most point on the referred plots.)

---

> > ### Author Response · Authors · 2024-11-22
> >
> > > all the benchmarks used in the results section are out of date and focus on easy tasks. The results would be much more compelling if the authors included a benchmark like MMLU. That being said, given the short rebuttal period I would put this experiment as a lower priority than the two above concerns
> >
> > We thank the reviewer for suggesting to include results on additional benchmarks. To further strengthen our work , we conduct extensive experiments on more recent benchmarks: MMMU, MMMU-Pro, VQAv2, and VizWiz for VLMs and MMLU and MMLU-Pro for LLMs. We show that our method can be applied to these challenging datasets and improve upon the zero-shot baseline. For instance, on the MMLU benchmark, unsupervised ICL brings 3% improvements upon the zero-shot and remarkably matches the performance of supervised ICL. Furthermore, on the challenging MMLU-Pro, our unsupervised ICL improves by 6% upon the zero-shot counterpart. We also observe similar trends on the newly added vision tasks. Indeed, our unsupervised ICL brings 2% absolute improvement upon the zero-shot baseline for the frontier GPT-4o model on the MMMU dataset. We kindly refer the reviewer to the general response for detailed results. Overall, these results highlight the strength of the proposed joint inference framework and its applicability to the frontier models and the most recent challenging benchmarks. We will include these results in the final version of the paper.
> >
> >
> > > Q2: are the in context examples used different than the questions the LLM is tasked to answer in a given call?
> >
> > The in-context examples are sampled randomly and it is ensured that they do not contain the question the LLM is tasked to answer in a given call. We will include these clarifications in the future revision of the manuscript.

---

> ### Comment · Reviewer_Nhh8 · 2024-11-25
> **Thank you**
>
> I thank the authors for addressing my concerns and adding a methods figure. With these new explanations and experiments I will raise my score to a 6. My one comment would be to please put results for the one iteration baseline for all the datasets in the final manuscript. I may have miscommunicated why i wanted this: I don't think the method is less interesting without this iteration step. In my experience with these iterative approaches they have very diminishing returns and 1 iteration is often similar or only slightly worse performance, and I was curious as to whether the authors saw this as well. From my impression of the authors reply, iterations do have diminishing returns, which I think would be very useful to know. If one iteration is just as good, why do more :)

---

> > ### Author Response · Authors · 2024-11-25
> >
> > We thank the reviewer for the response and for increasing the score! We are happy that our rebuttal has addressed reviewers' concerns and we are also grateful for the insightful feedback to improve the narration of our methodology. We will provide per-dataset convergence plots showing performance as a function of number of iterations for our unsupervised ICL in the final version of our manuscript.

---

### Official Review · Reviewer_UqwZ · 2024-11-03

**Soundness:** 2
**Presentation:** 2
**Contribution:** 3
**Rating:** 5
**Confidence:** 3

**Summary:**

The paper proposes an 'unsupervised joint inference' framework to improve the performance of large language / vision-language models on tasks without using labeled data. The proposed approach predicts output sequences simultaneously for multiple inputs, leveraging inter-sample dependencies. There are two variants in the proposed approach: 1. use the model's own predictions to finetune its parameters. 2. Use in-context learning to refine predictions iteratively. The paper conducts experiments on some nlp and vision-language datasets and claim substantial improvements to vanilla zero-shot approach and often on-par performance with supervised approaches.

**Strengths:**

1. The paper proposes an novel framework to adapt llm or vlm for unseen tasks.
2. The paper intents to build solid theoretical foundation for the proposed approach. It establishes a theoretical connection between the proposed unsupervised ICL method and the joint inference objective, showing how the former approximates the latter.
3. The experiments conducted are diverse on different tasks: text classification, image classification, question answering, visual question answering, natural language inference, common-sense reasoning, and math problem-solving.
4. Experiments show that the proposed approach is effective and outperform the baselines by a large margin.

**Weaknesses:**

1. The paper overclaims it's contribution and effectiveness of its approach. In abstract, the paper claims that 'our approach, although unsupervised, often performs on par with supervised approaches that use ground truth labels.' However, its an overclaim for the following reasons: (1) In the results shown in the paper, most of the time supervised approach is better than the proposed approach. (2) Supervised approach uses LoRA instead of full model training.
2. The experiments are not solid. Some of the experiments in the paper use non-standard (potentially easy/small/old/deprecated) datasets. Most notably - CIFAR 10/100 for image classification and COCO-colar and COCO-number for VQA. The results are much less convincing compared to ImageNet for classification and VQA v2 for VQA.

**Questions:**

Is there a reason for picking these datasets for experiments? Is it possible to perform the same experiments on more recent and broadly adopted datasets?

---

> ### Author Response · Authors · 2024-11-22
>
> We thank the reviewer for the valuable feedback and for acknowledging the novelty of our joint inference framework. We are glad that the reviewer recognizes the solid theoretical foundation for the proposed approaches. We are also happy that the reviewer appreciates the diversity of the provided experimental study that covers various tasks and models, highlighting the effectiveness of the proposed approach. Below, we address the reviewer’s concerns and we hope that the provided clarifications and the additional experiments help to increase the reviewer’s confidence in our work.
>
> > The paper overclaims it's contribution and effectiveness of its approach. In abstract, the paper claims that 'our approach, although unsupervised, often performs on par with supervised approaches that use ground truth labels.' However, its an overclaim for the following reasons: (1) In the results shown in the paper, most of the time supervised approach is better than the proposed approach. (2) Supervised approach uses LoRA instead of full model training.
>
> We thank the reviewer for their suggestion to improve the presentation of our contributions. We would like to note that, as Table 1 shows, we find that for 10 out of 13 considered NLP, unsupervised ICL closes at least 85% of the relative gap, i.e., (uICL - ZS) / (supICL - ZS),  between zero-shot and supervised ICL. It is also important to note that supervised ICL uses ground truth examples to perform adaptation to a task, thus it stands as an upper bound for our unsupervised ICL. We believe that our results provide a remarkable insight into how close unsupervised method can come to the supervised counterpart. We will edit the abstract in the future revision of our manuscript to reflect found trends more objectively.
>
> Our unsupervised fine-tuning approach can be combined with any fine-tuning strategy. In response to the reviewer’s request to include full model training and to  further support usage of LoRA fine-tuning instead of full model training, we provide the ablation of the task encoder’s design and its effect on the performance of the unsupervised fine-tuning approach. We consider the Llama 3.1-8B foundation model and the SST2, RTE and Boolq datasets as illustrative tasks. We consider three types of the task encoder: (i) LoRA parameter efficient fine-tuning of the foundation model as done by default in the paper, (ii) Full fine-tuning of the entire foundation model, (iii) linear model on top of frozen Llama embeddings of the input instance.
>
> |                 | Trainable Param. | sst2 |  rte | boolq |
> |-----------------|:----------------:|:----:|:----:|:-----:|
> | Zero-Shot       |         -        | 77.7 | 62.7 |  66.7 |
> | Unsupervised FT |       LoRA       | 92.3 | 81.7 |  81.7 |
> |                 |      Linear      | 89.6 | 72.8 |  78.6 |
> |                 |        Full      | 92.3 | 80.3 |  81.3 |
> | Supervised FT   |       LoRA       | 92.1 | 89.0 |  85.6 |
> |                 |      Linear      | 90.1 | 74.4 |  79.9 |
> |                 |        Full      | 92.0 | 82.4 |  85.7 |
>
> First, unsupervised fine-tuning with all three task encoders substantially outperforms the zero-shot performance on each of the considered datasets. For instance, even with the simple linear task encoder, unsupervised fine-tuning improves by 12%, 10% and 12% on the SST2, RTE and Boolq dataset respectively. On the other hand, we can observe that linear task encoder significantly underfits both LoRA and full fine-tuning. In particular, LoRA fine-tuning outperforms the linear task encoder by 3%, 9% and 3% on the SST2, RTE and Boolq datasets respectively.
>
> Furthermore, it can be observed that supervised fine-tuning with full model training severely overfits on the RTE dataset compared to LoRA parameter efficient fine-tuning. In turn, the unsupervised fine-tuning with full model training inherits this problem and performs worse compared to the LoRA fine-tuning. In particular, compared to LoRA supervised fine-tuning, full supervised fine-tuning drops by 7% on the RTE dataset. This, in turn, results in a 2% drop of the performance of full unsupervised fine-tuning compared to LoRA unsupervised fine-tuning.
>
> Overall, the obtained results suggest that the LoRA fine-tuning provides the practical trade-off between parameter efficiency and the performance. We will include these ablations in the future revision of our manuscript.

---

> > ### Author Response · Authors · 2024-11-22
> >
> > > The experiments are not solid. Some of the experiments in the paper use non-standard (potentially easy/small/old/deprecated) datasets. Most notably - CIFAR 10/100 for image classification and COCO-colar and COCO-number for VQA. The results are much less convincing compared to ImageNet for classification and VQA v2 for VQA.
> >
> > &
> >
> > > Q1: Is there a reason for picking these datasets for experiments? Is it possible to perform the same experiments on more recent and broadly adopted datasets?
> >
> > We thank the reviewer for suggesting to include results on additional benchmarks. First, as shown in Table 3 in our main paper, we would like to note that VLMs are still struggling on the standard problems such as image classification. For instance, the zero-shot performance of OpenFlamingo on the fine-grained Food101 dataset included in our work is 58.4%, while the CLIP zero-shot model, that is used as a vision encoder in OpenFlamingo, achieves 92.9% [1]. The similar trend is observed on the CIFAR100 dataset, where OpenFlamingo zero-shot achieves 58% and CLIP achieves 77.9%. These results highlight the necessity of evaluation and improvement of VLMs on the standard image classification problem. Furthermore, we want to pinpoint that we provide ImageNet-100 results with GPT-4o in Table 4 in the main paper. These results show that our unsupervised ICL is able to provide significant improvement over the zero-shot baseline even for the frontier GPT-4o model. In particular, it brings 3% absolute improvement and closely approaches supervised ICL.
> >
> > To further strengthen our work , we conduct extensive experiments on more recent benchmarks: MMMU, MMMU-Pro, VQAv2, and VizWiz for VLMs and MMLU and MMLU-Pro for LLMs. We kindly refer the reviewer to the general response for detailed results. We show that our method can be applied to these challenging datasets and improve upon the zero-shot baseline. For instance, on the MMLU benchmark, unsupervised ICL brings 3% improvements upon the zero-shot and remarkably matches the performance of supervised ICL. Furthermore, on the challenging MMLU-Pro, our unsupervised ICL improves by 6% upon the zero-shot counterpart. We also observe similar trends on the newly added vision tasks. Indeed, our unsupervised ICL brings 2% absolute improvement upon the zero-shot baseline for the frontier GPT-4o mode on the MMMU dataset. Moreover, on the challenging VizWiz dataset, our unsupervised ICL achieves 5% absolute improvement upon the zero-shot baseline. Overall, these results highlight the strength of the proposed joint inference framework and its applicability to the frontier models and the most recent challenging benchmarks.
> >
> > [1] Radford et al. Learning Transferable Visual Models from Natural Language Supervision. ICML 2021.

---

### Official Review · Reviewer_UFdR · 2024-11-04

**Soundness:** 3
**Presentation:** 3
**Contribution:** 3
**Rating:** 6
**Confidence:** 4

**Summary:**

This paper presents a joint inference framework for large language and vision-language models, with an optimized and efficient approximation techniques to enable unsupervised adaptation to new tasks through two main methods: unsupervised fine-tuning and unsupervised in-context learning (ICL). This approach makes predictions jointly across multiple inputs, and shows significant improvements over zero-shot inference, sometimes achieving results close to supervised approaches without needing labeled data.

**Strengths:**

1) A big strength of this method is that it provides a framework to effectively allows unsupervised fine-tuning and ICL, reducing the need for labeled examples which is usually expansive in practice.
2) It also demonstrates considerable accuracy and generalization improvements across a range of language and vision tasks, even comparing with supervised fine-tuning methods. The experiments performed on multiple datasets, models, and tasks, demonstrating the framework’s effectiveness in both text and vision-language scenarios.
3) It generally works for various model types such as open-weight models (e.g., Llama-3.1, OpenFlamingo) and close-weight models (e.g., GPT-4), showing flexibility and robustness in application.

**Weaknesses:**

1) The effectiveness of unsupervised ICL seems to rely on the model’s inherent in-context learning abilities. This may restrict its applicability to models lacking strong ICL capabilities. It is unclear how does the same method work for other models with relative weak capability.
2) While the experimental results are promising, a deeper theoretical analysis of the joint inference framework’s optimization behavior and potential limitations or failures would strengthen the contribution.

**Questions:**

Q1: Do you have any evaluations on how the joint inference framework affects model robustness, especially when inputs are noisy or out-of-distribution?

Q2: It is also very interesting to see if there are any common failure cases/scenarios of the proposed method?

---

> ### Author Response · Authors · 2024-11-22
>
> We thank the reviewer for the positive evaluation of our work and for acknowledging the importance of our framework that enables unsupervised adaptation of foundation models, reducing the need for labeled examples that are expensive to obtain in practice. We are glad that the reviewer recognizes the substantial improvements brought by the proposed approaches, even when compared to the corresponding supervised baselines. We are also glad that the reviewer appreciates the diversity of the provided experiments that demonstrate the effectiveness of our framework in both text and vision-language scenarios. Finally, we thank the reviewer for acknowledging the flexibility and robustness of our framework that is applicable to both open-weight and close-weight models. Below, we provide clarifications to the reviewer’s concerns and we hope that our response helps to improve the reviewer’s confidence in our work.
>
> > The effectiveness of unsupervised ICL seems to rely on the model’s inherent in-context learning abilities. This may restrict its applicability to models lacking strong ICL capabilities. It is unclear how does the same method work for other models with relative weak capability.
>
> We thank the reviewer for the comment. In general, smaller models are indeed expected to have weaker ICL capabilities resulting in smaller improvements of the supervised ICL compared to the zero-shot baseline. To  further strengthen our experimental evaluation, we evaluate performance of using smaller models and conduct extensive experiments on RTE, SST2 and BoolQ datasets (tables below). As expected, we observe smaller improvements of the supervised ICL compared to the zero-shot baseline. For example, on the RTE dataset, supervised ICL for 1B model brings 11% relative improvement upon the corresponding zero-shot baseline, while supervised ICL for 3B and 7B models bring 22% and 29% relative improvements upon the corresponding zero-shot baselines respectively. Nevertheless, we can observe that even much smaller models can exhibit reasonable in-context learning abilities, and our unsupervised ICL is also applicable to them in that scenario. We also kindly refer the reviewer to the general response for the detailed discussion of reliance of our joint inference framework on the in-context learning capabilities of a foundation model.
>
>
> |     RTE         | 0-shot | unsupervised ICL | supervised ICL |
> |--------------|--------|------------------|----------------|
> | Llama 3.1 8B | 62.7   | 78.8             | 80.8           |
> | Llama 3.2 3B | 61.6   | 73.7             | 75.0           |
> | Phi 1.5      | 60.3   | 60.5             | 61.8           |
> | Llama 3.2 1B | 50.8   | 55.8             | 56.5           |
>
>
> |    SST2          | 0-shot | unsupervised ICL | supervised ICL |
> |--------------|--------|------------------|----------------|
> | Llama 3.1 8B | 77.7   | 92.4             | 93.3           |
> | Llama 3.2 3B | 72.4   | 93.3             | 92.5           |
> | Phi 1.5      | 58.5   | 84.7             | 83.5           |
> | Llama 3.2 1B | 68.6   | 54.3             | 61.0           |
>
>
> |    BoolQ          | 0-shot | unsupervised ICL | supervised ICL |
> |--------------|--------|------------------|----------------|
> | Llama 3.1 8B | 66.7   | 82.6             | 84.1           |
> | Llama 3.2 3B | 51.3   | 75.1             | 76.2           |
> | Phi 1.5      | 59.7   | 57.3             | 56.4           |
> | Llama 3.2 1B | 50.0   | 50.0             | 54.9           |

---

> > ### Author Response · Authors · 2024-11-22
> >
> > > Q2: It is also very interesting to see if there are any common failure cases/scenarios of the proposed method?
> >
> > As we also discuss in our general response, our framework relies on the ICL abilities of a foundation model in the same fashion as Chain-of-Thought prompting relies on the inherent reasoning abilities of a foundation model. As soon as foundation models acquire these abilities, our joint inference framework will be readily available to improve them even further without any additional supervision. Our experiments in response to the previous question show that even small foundation models can exhibit ICL abilities. Subsequently, our unsupervised ICL already improves upon the zero-shot baseline for such models.
> >
> > Although some models might exhibit weak ICL abilities, they still can be improved using our unsupervised fine-tuning approach. We can employ larger models with better ICL abilities to improve the smaller one. In particular, the joint inference objective (Eq. 8) is computed using a larger model to provide feedback to a smaller model. Below, we provide the results of such experiment with the small Llama 3.2 1B as our task encoder, and we consider 1B, 8B and 70B models to compute the joint inference objective.
> >
> > | Method     | Objective computed with| SST2 | RTE  | BoolQ |
> > |-|-|--|--|--|
> > | Zero-shot | | 68.6 | 50.8 | 50.0 |
> > | Unsup FT   | Llama 3.2 1B | 90.5 | 60.8 | 57.4 |
> > |   | Llama 3.2 8B  | 90.8 | 74.5 | 73.4  |
> > |  | Llama 3.2 70B-Instruct | 90.6 | 75.6 | 74.4 |
> > | Sup FT     | -   | 90.0 | 79.3 | 75.5 |
> >
> > First, it is easy to see that even using the same 1B model to compute the joint inference objective leads to substantial gains in the performance (table above). For example, as shown in the experiment of the previous response, supervised ICL on the RTE dataset leads to only 5% improvement upon the zero-shot baseline. In turn, unsupervised fine-tuning leads to 10% improvement, outperforming the supervised ICL (table above).
> >
> > Moreover,  switching to larger models for computing the joint inference objective boosts the performance of the small Llama 3.2 1B model even further, closely approaching the supervised fine-tuning upper-bound. For instance, on the BoolQ dataset, using the 70B model in the objective results in remarkable 24% absolute gains over the zero-shot baseline. We thank the reviewer again for the insightful suggestion to perform the additional analysis and we will include it in the future revision of our manuscript.

---

> > > ### Author Response · Authors · 2024-11-22
> > >
> > > > While the experimental results are promising, a deeper theoretical analysis of the joint inference framework’s optimization behavior and potential limitations or failures would strengthen the contribution.
> > >
> > > We thank the reviewer for their interest in a deeper understanding of our proposed methods. Below we clarify the theoretical foundations of both our proposed methods that we provide in our work.
> > >
> > > Both our proposed methods, unsupervised ICL and unsupervised fine-tuning (FT), are developed as optimization methods that approximate the optimization of the joint inference objective in Eq. (8). As also recognized by reviewer UqwZ, we provide the following theoretical foundations for these approximate methods in our work:
> > > * In Sec. 4.2, the unsupervised FT method is derived as a principled approach to optimizing the joint inference objective. Specifically, Eq (9) shows that our unsupervised FT method optimizes the lower bound of the joint objective.
> > > * In Sec 4.3, we show that the unsupervised ICL method can be viewed as an optimization method through iterative sampling. In Fig. 2, we, indeed, demonstrate that more optimization steps of the unsupervised ICL method lead to a higher joint inference objective and, subsequently, higher performance.
> > >
> > > We will extend the discussion of our method to include these clarifications in our final manuscript. In addition, we discussed the potential limitations and failure modes in our response to the previous questions.
> > >
> > > > Q1: Do you have any evaluations on how the joint inference framework affects model robustness, especially when inputs are noisy or out-of-distribution?
> > >
> > > We kindly refer the reviewer to the general response for the new obtained results on very challenging benchmarks such as MMLU, MMLU-Pro, MMMU and MMMU-Pro. These benchmarks contain diverse question answering tasks covering a wide range of topics and are not part of the foundation model pre-training pipeline. Our results suggest that our joint inference framework is also applicable on these benchmarks and improves the performance upon the zero-shot baseline. For instance, on the MMLU benchmark, unsupervised ICL brings 3% improvements upon the zero-shot and remarkably matches the performance of supervised ICL. Furthermore, on the challenging MMLU-Pro, our unsupervised ICL improves by 6% upon the zero-shot counterpart. We also observe similar trends on the newly added vision tasks. Indeed, our unsupervised ICL brings 2% absolute improvement upon the zero-shot baseline for the frontier GPT-4o model on the MMMU dataset. Overall, these results highlight the strength of the proposed joint inference framework and its applicability to the frontier models and the most recent challenging benchmarks.

---

> > > > ### Comment · Reviewer_UFdR · 2024-12-02
> > > >
> > > > Thank you to authors for the detailed responses. My concerns have been well addressed. After carefully reviewing the other reviews and responses, I believe the manuscript is well-suited for acceptance. Therefore, I am maintaining my original positive rating.

---

> > > > > ### Author Response · Authors · 2024-12-02
> > > > >
> > > > > We thank the reviewer for the response and appreciate the reviewer's feedback. We are happy that our rebuttal has well addressed reviewers' concerns and we are also glad that the reviewer believes our manuscript is well-suited for the acceptance.

---

### Author Response · Authors · 2024-11-22
**General Response**

We thank all the reviewers for their time and provided feedback. We are glad that the reviewers recognize the importance of enabling effective unsupervised adaptation of foundation models (UFdR, UqwZ, TGs1), diversity of our experimental study (UFdR, UqwZ, TGs1), broad applicability (UFdR, Nhh8) and significance of improvements brought by our framework (UFdR, Nhh8, TGs1).

We respond to each reviewer individually. Below, we summarize additional experiments added during the rebuttal and answer a common question about the dependence on ICL capabilities raised by reviewers UFdR and TGs1.

# Additional Results:
We add the following new results for the rebuttal, which will also be included in the manuscript:
- We evaluate our methods on 6 recent challenging benchmarks: MMLU, MMLU-Pro for LLMs, and MMMU, MMMU-Pro, VQAv2, and VizWizz for VLMs (see Reb-Tab. 1-3 below.) We show that our method is applicable to these benchmarks and improves upon zero-shot inference.
- We provide additional results of our methods on 3 small models: Llama-3.2-1B/3B and Phi-1.5. We find that our methods can be applied to a broad range of model sizes and improve upon their zero-shot performance (see additional discussion below.)
- We explore different task encoder parameterizations for the unsupervised FT method. We show that our method can be applied to any parametrization, with LoRA resulting in the best performance.


| Method| MMLU | MMLU-Pro |
|-|-|-|
| Zero-shot| 62.6 | 26.1|
| Unsupervised ICL| 65.4 | 32.5|
| Supervised ICL| 65.4 | 37.2|

__Reb-Tab.1: MMLU and MMLU-pro with Llama-3.1-8B__ We use CoT for MMLU-Pro.

|Method|MMMU|MMMU-Pro|
|--|-|-|
|Zero-shot|66.4|54.7|
|Unsupervised ICL|68.6|55.5|

__Reb-Tab.2: MMMU and MMMU-Pro with GPT-4o.__ We use CoT. Note that supervised ICL is not possible as no ground truth reasonings is available for these datasets to perform CoT.

|Method| VQAv2 | VizWiz |
|-|-|-|
|Zero-shot| 58.1|41.6|
|Unsupervised ICL| 59.7| 46.7|
|Supervised ICL| 60.6| 55.8|

__Reb-Tab.3: VQAv2 and VizWiz with OpenFlamingo-9B__.
# Reliance on the Model’s ICL capabilities
First, we define ICL capabilities to support the discussion.
__ICL Capabilities definition:__ A model exhibits “ICL capabilities” if its few-shot ICL performance is higher than its 0-shot performance.

Indeed, in order for our joint inference framework (via unsupervised ICL or unsupervised fine-tuning) to improve the model's 0-shot performance, the underlying model should exhibit ICL capabilities in the first place. In these cases, our approach allows one to invoke the model’s ICL capability and improve upon 0-shot without the need for ground truth labels. This is similar to how, for example, a widely adopted CoT method improves zero-shot performance relying on the models’ reasoning capabilities and doesn’t work otherwise [1, 2]. We believe that our proposed methods are of practical value given recent improvements in models’ ICL capabilities, which we briefly discuss below.

Most recent language-only models, indeed, exhibit ICL capabilities [6]. In our experiments, we find that even small models such as Llama-3B or Phi-1.5 can exhibit ICL capabilities improving upon zero-shot in many cases. In these cases, we find that our unsupervised ICL method also improves zero-shot performance (see Tab. 1, Fig. 3, and the table in response to TGs1). Even in cases when a small model doesn’t exhibit ICL capabilities, we show that our unsupervised fine-tuning method can be used to improve this small model using feedback from a larger model with stronger ICL capabilities (see the table in response to TGs1). This makes both our methods readily applicable to a broad range of LLMs.

Enabling ICL capabilities in vision-language models, on the other hand, is still an ongoing research direction (e.g., [4, 5]). We find that open-source models might not exhibit ICL capabilities on more recent benchmarks (e.g., LLava-Interleave on MMMU), an observation also made in [3]. However, for more capable closed-source and/or larger models such as GPT-4o, which do exhibit ICL capabilities, we show that our unsupervised ICL method is already applicable and improves upon 0-shot performance  (see Tab. 3 and Reb-Tab. 2-3). This demonstrates that our method is already applicable to existing VLMs with ICL capabilities and will be readily available for a broader range of models when they acquire this capability.


[1]Chain-of-Thought Prompting Elicits Reasoning in Large Language Models. Wei et al. NeurIPS 2022

[2]Large Language Models are Zero-Shot Reasoners. Kojima et al. NeurIPS 2022

[3]The Devil in the Details of Benchmarking Multimodal In-Context Learning. Zong et al., arXiv preprint

[4]MM1: methods, analysis and insights from multimodal LLM pre-training. McKinzie et al., ECCV 2024

[5]MMICL: Empowering vision-language model with multi-modal in-context learning. Zhao et al., ICLR 2024

[6]Language models are few-shot learners. Brown et al., arXiv preprint, 2020

---

### Meta-Review · Area_Chair_i47k · 2024-12-20

**Metareview:**

**Summary:**

This paper proposes an unsupervised joint inference framework for large language models and vision-language models. This approach performs unsupervised adaptation through unsupervised fine-tuning and unsupervised in-context learning, leveraging inter-sample dependencies. Experimental results demonstrate that the proposed method substantially improves performance of language models (LLaMA-3 and GPT-4o) and a vision-language model (OpenFlamingo), compared to zero-shot models and achieves performance comparable to supervised approaches.

**Strengths:**

1. **Underexplored unsupervised adaptation of LLM and VLM.** This paper studies an underexplored topic, unsupervised adaptation of LLM and VLM using unsupervised fine-tuning and in-context learning. The proposed method iteratively improve the models predication at test time.
2. **Strong experimental results.** The proposed method significantly improves the performance compared to zero-shot predictions.
3. **Generalizability.** The proposed method is effective for both LLM and VLM, including open and close-weight models. The authors demonstrated the effectiveness across a wide range of tasks: text classification, image classification, question answering, visual question answering, natural language inference, common-sense reasoning, and math problem-solving.

**Weaknesses:**

1. **Dependence on in-context learning capability.** Unsupervised ICL heavily relies on the strong ICL capabilities of the pre-trained models.
2. **Suboptimal task encoder.** More task encoders could be considered rather than LoRA.
3. **Presentation.** As Reviewer Nhh8 pointed out, the clarity of this paper needs to be improved.

**Main reasons:**

This paper studies an underexplored problem, which is the unsupervised adaptation of pre-trained LLM and VLM. It is an important problem and the proposed method achieves strong performance and it is applicable to a wide range of tasks and models, including open and closed-weights.

**Additional Comments On Reviewer Discussion:**

The reviewers asked questions and most concerns are addressed by the authors feedback.
1. Necessity of the iterative methods. The authors provided the comparison between one-iteration and multi-iteration methods.
2. Computational cost. The efficiency/FLOPs has been analyzed during rebuttal.
3. Additional experimental results support that the proposed method is applicable/effective to open-ended tasks.

---

### Decision · Program_Chairs · 2025-01-22

Accept (Poster)